# TimeAutoML: Autonomous Representation Learning for Multivariate Irregularly Sampled Time Series

## Abstract

Multivariate time series (MTS) data are becoming increasingly ubiquitous in diverse domains, e.g., IoT systems, health informatics, and 5G networks. To obtain an effective representation of MTS data, it is not only essential to consider unpredictable dynamics and highly variable lengths of these data but also important to address the irregularities in the sampling rates of MTS. Existing parametric approaches rely on manual hyperparameter tuning and may cost a huge amount of labor effort. Therefore, it is desirable to learn the representation automatically and efficiently. To this end, we propose an autonomous representation learning approach for multivariate time series (TimeAutoML) with irregular sampling rates and variable lengths. As opposed to previous works, we first present a representation learning pipeline in which the configuration and hyperparameter optimization are fully automatic and can be tailored for various tasks, e.g., anomaly detection, clustering, etc. Next, a negative sample generation approach and an auxiliary classification task are developed and integrated within TimeAutoML to enhance its representation capability. Extensive empirical studies on real-world datasets demonstrate that the proposed TimeAutoML outperforms competing approaches on various tasks by a large margin. In fact, it achieves the best anomaly detection performance among all comparison algorithms on 78 out of all 85 UCR datasets, acquiring up to 20% performance improvement in terms of AUC score.

## 1 Introduction

The past decade has witnessed a rising proliferation in Multivariate Time Series (MTS) data, along with a plethora of applications in domains as diverse as IoT data analysis, medical informatics, and network security. Given the huge amount of MTS data, it is crucial to learn their representations effectively so as to facilitate underlying applications such as clustering and anomaly detection. For this purpose, different types of methods have been developed to represent time series data.

Traditional time series representation techniques, e.g., Discrete Fourier Transform (DCT) (Faloutsos et al., 1994), Discrete Wavelet Transform (DWT)(Chan & Fu, 1999), Piecewise Aggregate Approximation (PAA)(Keogh et al., 2001), etc., represent raw time series data based on specific domain knowledge/data properties and hence could be suboptimal for subsequent tasks given the fact that their objectives and feature extraction are decoupled.

More recent time series representation approaches, e.g., Deep Temporal Clustering Representation (DTCR) (Ma et al., 2019), Self-Organizing Map based Variational Auto Encoder (SOM-VAE) (Fortuin et al., 2018), etc., optimize the representation and the underlying task such as clustering in an end-to-end manner. These methods usually assume that time series under investigation are uniformly sampled with a fixed interval. This assumption, however, does not always hold in many applications. For example, within a multimodal IoT system, the sampling rates could vary for different types of sensors.

Unsupervised representation learning for irregularly sampled multivariate time series is a challenging task and there are several major hurdles preventing us from building effective models: i) the design of neural network architecture often employs a trial and error procedure which is time consuming and could cost a substantial amount of labor effort; ii) the irregularity in the sampling rates

constitutes a major challenge against effective learning of time series representations and render most existing methods not directly applicable; iii) traditional unsupervised time series representation learning approach does not consider contrastive loss functions and consequently only can achieve suboptimal performance.

To tackle the aforementioned challenges, we propose an autonomous unsupervised representation learning approach for multivariate time series to represent irregularly sampled multivariate time series. TimeAutoML differs from traditional time series representation approaches in three aspects. First, the representation learning pipeline configuration and hyperparameter optimization are carried out automatically. Second, a negative sample generation approach is proposed to generate negative samples for contrastive learning. Finally, an auxiliary classification task is developed to distinguish normal time series from negative samples. In this way, the representation capability of TimeAutoML is greatly enhanced. We conduct extensive experiments on UCR time series datasets and UEA multivariate time series datasets. Our experiments demonstrate that the proposed TimeAutoML outperforms comparison algorithms on both clustering and anomaly detection tasks by a large margin, especially when time series data is irregularly sampled.

## 2 RELATED WORK

**Unsupervised Time Series Representation Learning**   Time series representation learning plays an essential role in a multitude of downstream analysis such as classification, clustering, anomaly detection. There is a growing interest in unsupervised time series representation learning, partially because no labels are required in the learning process, which suits very well many practical applications. Unsupervised time series representation learning can be broadly divided into two categories, namely 1) multi-stage methods and 2) end-to-end methods. Multi-stage methods first learn a distance metric from a set of time series, or extract the features from the time series, and then perform downstream machine learning tasks based on the learned or the extracted features. Euclidean distance (ED) and Dynamic Time Warping (DTW) are the most commonly used traditional time series distance metrics. Although the ED is competitive, it is very sensitive to outliers in the time series. The main drawback of DTW is its heavy computational burden. Traditional time series feature extraction methods include Singular Value Decomposition (SVD), Symbolic Aggregate Approximation (SAX), Discrete Wavelet Transform (DWT)(Chan & Fu, 1999), Piecewise Aggregate Approximation (PAA)(Keogh et al., 2001), etc. Nevertheless, most of these traditional methods are for regularly sampled time series, so they may not perform well on irregularly sampled time series. In recent years, many new feature extraction methods and distance metrics are proposed to overcome the drawbacks mentioned above. For instance, Paparrizos & Gravano (2015); Petitjean et al. (2011) combine the proposed distance metrics and K-Means algorithm to achieve clustering. Lei et al. (2017) first extracts sparse features of time series, which is not sensitive to outliers and irregular sampling rate, and then carries out the K-Means clustering. In contrast, end-to-end approaches learn the representation of the time series in an end-to-end manner without explicit feature extraction or distance learning (Fortuin et al., 2018; Ma et al., 2019). However, the aforementioned methods need to manually design the network architecture based on human experience which is time-consuming and costly. Instead, we propose in this paper a representation learning method which optimizes an AutoML pipeline and their hyperparameters in a fully autonomous manner. Furthermore, we consider negative sampling and contrastive learning in the proposed framework to effectively enhance the representation ability of the proposed neural network architecture.

**Irregularly Sampled Time Series Learning**   There exist two main groups of works regarding machine learning for irregularly sampled time series data. The first type of methods impute the missing values before conducting the subsequent machine learning tasks (Shukla & Marlin, 2019; Luo et al., 2018; 2019; Kim & Chi, 2018). The second type directly learns from the irregularly sampled time series. For instance, Che et al. (2018); Cao et al. (2018) propose a memory decay mechanism, which replaces the memory cell of RNN by the memory of the previous timestamp multiplied by a learnable decay coefficient when there are no sampling value at this timestamp. Rubanova et al. (2019) combines RNN with ordinary differential equation to model the dynamic of irregularly sampled time series. Different from the previous works, TimeAutoML makes use of the special characteristics of RNN (Abid & Zou, 2018) and automatically configure a representation learning pipeline to model the temporal dynamics of time series. It is worthy mentioning that there are many types of

irregularly sampled time series, which may be caused by sensor failure or sampling time error. And what we put emphasis on analyzing in this paper is a special type of irregularly sampled time series, which have many missing timestamps compared to regularly sampled time series.

**AutoML** Automatic Machine Learning (AutoML) aims to automate the time-consuming model development process and has received significant amount of research interests recently. Previous works about AutoML mostly emphasize on the domains of computer vision and natural language processing, including object detection (Ghiasi et al., 2019; Xu et al., 2019; Chen et al.), semantic segmentation (Weng et al., 2019; Nekrasov et al., 2019; Bae et al., 2019), translation (Fan et al., 2020) and sequence labeling (Chen et al., 2018a). However, AutoML for time series learning is an underappreciated topic so far and the existing works mainly focus on supervised learning tasks, e.g., time series classification. Ukil & Bandyopadhyay propose an AutoML pipeline for automatic feature extraction and feature selection for time series classification. Van Kuppevelt et al. (2020) develops an AutoML framework for supervised time series classification, which involves both neural architecture search and hyperparameter optimization. Olsavszky et al. (2020) proposes a framework called AutoTS, which performs time series forecasting of multiple diseases. Nevertheless, to our best knowledge, no previous work has addressed unsupervised time series learning based on AutoML.

**Summary of comparisons with related work** We next provide a comprehensive comparison between the proposed framework and other state-of-the-art methods, including (WaRTEm (Mathew et al., 2019), DTCR (Ma et al., 2019), USRLT (Franceschi et al., 2019) and BeatGAN (Zhou et al., 2019)), as shown in Table 1. In particular, we emphasize on a total of seven features in the comparison, including data augmentation, negative sample generation, contrastive learning, selection of autoencoders, similarity metric selection, attention mechanism selection, and automatic hyperparameter search. TimeAutoML is the only method that has all the desired properties.

Table 1: Comparisons with related methods

|  | WaRTEm | DTCR | USRLT | BeatGAN | TimeAutoML |
|---|---|---|---|---|---|
| Data augmentation |  |  |  | ✓ | ✓ |
| Negative sample generation | ✓ | ✓ | ✓ |  | ✓ |
| Contrastive training | ✓ | ✓ | ✓ |  | ✓ |
| Autoencoder selection |  |  |  |  | ✓ |
| Similarity metric selection |  |  |  |  | ✓ |
| Attention mechanism selection |  |  |  |  | ✓ |
| Automatic hyperparameter search |  |  |  |  | ✓ |

## 3 TimeAutoML Framework

### 3.1 Proposed AutoML Framework

Let $\boldsymbol{X} = \{\underline{\boldsymbol{x}}_1, \underline{\boldsymbol{x}}_2, \cdots \underline{\boldsymbol{x}}_N\}$ denote a set of $N$ time series in which $\underline{\boldsymbol{x}}_i \in \mathbb{R}^{T_i}$, where $T_i$ is the length of the $i^{th}$ time series $\underline{\boldsymbol{x}}_i$. We aim to build an automated time series representation learning framework to generate task-aware representations that can support a variety of downstream machine learning tasks. In addition, we consider negative sample generation and contrastive self-supervised learning. The contrastive loss function focuses on building time series representations by learning to encode what makes two time series similar or different. The proposed TimeAutoML framework can automatically configure an representation learning pipeline with an array of functional modules, each of these modules is associated with a set of hyperparameters. We assume there are a total of $M$ modules and there are $Q_i$ options for the $i^{th}$ functional module. Let $\underline{\boldsymbol{k}}_i \in \{0,1\}^{Q_i}$ denote an indicating vector for $i^{th}$ module, with the constraint $1^\top \underline{\boldsymbol{k}}_i = \sum_{j=1}^{Q_i} k_{i,j} = 1$ ensuring that only a single option is chosen for each module. Let $\underline{\boldsymbol{\theta}}_{i,j}$ be the hyperparameters of $j^{th}$ option in $i^{th}$ module, where $\underline{\boldsymbol{\theta}}_{i,j}^C$ and $\underline{\boldsymbol{\theta}}_{i,j}^D$ are respectively the continuous and discrete hyperparameters. Let $\Theta$ and K denote the set of variables to optimize, i.e., $\Theta = \{\underline{\boldsymbol{\theta}}_{i,j}, \forall i \in [M], j \in [Q_i]\}$ and $K = \{\underline{\boldsymbol{k}}_1, \ldots, \underline{\boldsymbol{k}}_M\}$. We further let $f(K, \Theta)$ denote the corresponding objective function value. Please note that the objective function differs for different tasks. For anomaly detection, we use Area Under the Receiver Operating Curve (AUC) as objective function while we use the Normalized Mutual Information (NMI) as objective function for

clustering. The optimization problem of automatic pipeline configuration is shown below.

$$
\max_{\mathrm{K},\Theta} f(\mathrm{K},\Theta)
$$
$$
\text{subject to} \left\{
\begin{array}{l}
\underline{\boldsymbol{k}}_i \in \{0,1\}^{Q_i}, 1^\top \underline{\boldsymbol{k}}_i = 1, \forall i \in [M], \\
\underline{\boldsymbol{\theta}}_{i,j}^C \in C_{i,j}, \underline{\boldsymbol{\theta}}_{i,j}^D \in D_{i,j}, \forall i \in [M], j \in [Q_i].
\end{array}
\right.
\tag{1}
$$

We solve problem (1) by alternatively leveraging Thompson sampling and Bayesian optimization, which will be discussed as follows.

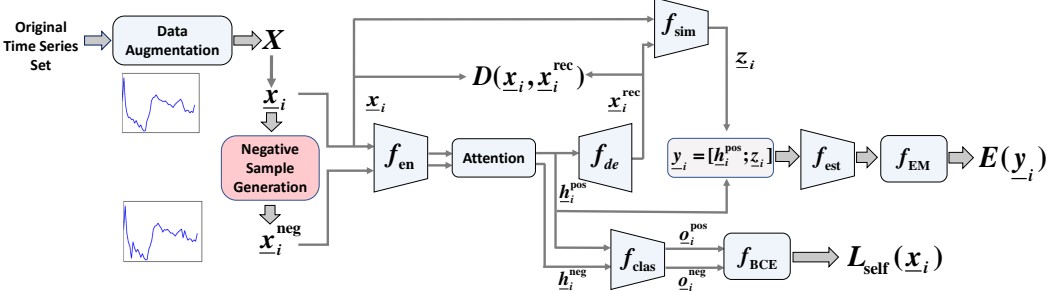

Figure 1: The representation learning pipeline of TimeAutoML. There are totally eight modules forming this pipeline, namely data augmentation, encoder $f_{\mathrm{en}}$, attention, decoder $f_{\mathrm{de}}$, similarity selection $f_{\mathrm{sim}}$, estimation network $f_{\mathrm{est}}$, EM estimator $f_{\mathrm{EM}}$, auxiliary classification network $f_{\mathrm{clas}}$. And $f_{\mathrm{BCE}}$ represents the binary cross entropy computation, $D(\underline{\boldsymbol{x}}_i, \underline{\boldsymbol{x}}_i^{\mathrm{rec}})$, $E(\underline{\boldsymbol{y}}_i)$ and $L_{\mathrm{self}}(\underline{\boldsymbol{x}}_i)$ represent the reconstruction loss, energy and self-supervised contrastive loss of input sample $\underline{\boldsymbol{x}}_i$, respectively.

### 3.1.1 PIPELINE CONFIGURATION

We first assume that the hyperparameters $\Theta$ are fixed during the pipeline configuration. We aim at selecting the better module option K to optimize objective function $f(\mathrm{K}, \Theta)$, we can delineate it as a $\mathbf{K} - \mathbf{max}$ problem:

$$
\mathrm{K} = \max_{\mathrm{K}} f(\mathrm{K}, \Theta) + \chi_K(\mathrm{K}), \quad \chi_K(\mathrm{K}) = \left\{
\begin{array}{ll}
0, & \text{if } \mathrm{K} \in K \\
-\infty, & \text{else}
\end{array}
\right. ,
\tag{2}
$$

where $K$ is the feasible set, i.e., $K = \{\mathrm{K} : \mathrm{K} = \{\underline{\boldsymbol{k}}_i\}, \underline{\boldsymbol{k}}_i \in \{0,1\}^{Q_i}, 1^\top \underline{\boldsymbol{k}}_i = 1, \forall i \in [M]\}$ and $\chi_K(\mathrm{K})$ is a penalty term that makes sure K fall in the feasible region.

Thompson sampling is utilized to tackle problem (2). In every iteration, Thompson sampling assumes the sampling probability of every option in each module follows Beta distribution, and the one corresponding to the maximum sampling value in each module will be chosen to construct the pipeline. After that, Beta distribution of the chosen options will be updated according to the performance of the configured pipeline. Due to space limitation, more details about Thompson sampling and the search space for pipeline configuration are shown in Appendix B and Appendix C, respectively.

The representation learning pipeline consists of eight modules, namely data augmentation, auxiliary classification network, encoder, attention, decoder, similarity selection, estimation network and EM estimator, as elucidated in Figure 1. The goal of data augmentation is to increase the diversity of samples. The auxiliary classification network aims at distinguishing the positive samples from generated negative samples, which will be discussed in detail in Section 3.2. And we combine encoder, attention, decoder and similarity selection together to generate the low-dimensional representation of the input time series. Given an input time series $\underline{\boldsymbol{x}}_i$, we can generate the latent space representation $\underline{\boldsymbol{y}}_i$, which is an concatenation of the output of $\underline{\boldsymbol{h}}_i$ and reconstruction error $\underline{\boldsymbol{z}}_i$, as shown below:

$$
\underline{\boldsymbol{h}}_i = f_{\mathrm{en}}(\underline{\boldsymbol{x}}_i), \quad \underline{\boldsymbol{x}}_i^{\mathrm{rec}} = f_{\mathrm{de}}(\underline{\boldsymbol{h}}_i), \quad \underline{\boldsymbol{z}}_i = f_{\mathrm{sim}}(\underline{\boldsymbol{x}}_i, \underline{\boldsymbol{x}}_i^{\mathrm{rec}}), \quad \underline{\boldsymbol{y}}_i = [\underline{\boldsymbol{h}}_i; \underline{\boldsymbol{z}}_i],
\tag{3}
$$

where $f_{\mathrm{en}}$ and $f_{\mathrm{de}}$ refer to an encoder and a decoder, respectively. There are three options for the encoder and decoder, namely, Recurrent Neural Network (RNN), Long Short Term Memory (LSTM),

Gated Recurrent Unit (GRU). $f_{\text{sim}}$ is a similarity function that characterizes the level of similarity between the original time series and the reconstructed one. Three possible similarity functions are considered in this paper, i.e., relative Euclidean distance, Cosine similarity, or concatenation of both.

After obtaining the latent space representation of the input time series, EM algorithm is then invoked to estimate the mean and convariance of GMM. Assuming there are $H$ mixture components in the GMM model, the mixture probability, mean, covariance for component $h$ in the GMM module can be expressed as $\phi_h, \underline{\boldsymbol{\mu}}_h, \sum_h$, respectively. Assuming there are a total of $N$ samples, the key parameters of GMM can be calculated as follows:

$$
\begin{aligned}
\underline{\boldsymbol{\gamma}}_i &= f_{\text{est}}(\underline{\boldsymbol{y}}_i), \forall i \in [N], \quad \phi_h = \sum_{i=1}^{N} \frac{\gamma_{i,h}}{N}, \forall h \in [H], \\
\underline{\boldsymbol{\mu}}_h, \sum_h &= f_{\text{EM}}(\{\underline{\boldsymbol{y}}_i, \underline{\boldsymbol{\gamma}}_{i,h}\}_{i=1}^{N}), \forall h \in [H],
\end{aligned}
\tag{4}
$$

where $f_{\text{est}}$ is the estimation network which is a multi-layer neural network, and $\underline{\boldsymbol{\gamma}}_i \in \mathbb{R}^H$ is the mixture-component membership prediction vector. $f_{\text{EM}}$ is the EM estimator which can estimate the mean and convariance of GMM via the EM algorithm. The $h^{th}$ entry of this vector represents the probability that $\underline{\boldsymbol{y}}_i$ belongs to the $h^{th}$ mixture component. The sample energy $E(\underline{\boldsymbol{y}}_i)$ is given by,

$$
E(\underline{\boldsymbol{y}}_i) = -\log\left(\sum_{h=1}^{H} \phi_h \cdot \frac{\exp(-\frac{1}{2}(\underline{\boldsymbol{y}}_i - \underline{\boldsymbol{\mu}}_h)^\top \sum_h^{-1} (\underline{\boldsymbol{y}}_i - \underline{\boldsymbol{\mu}}_h))}{\sqrt{|2\pi\sum_h|}}\right).
\tag{5}
$$

The sample energy can be used to characterize the level of anomaly of an input time series, that is, the sample with high energy will be deemed as an unusual time series. It is worth noticing that TimeAutoML may suffer from the *singularity* problem as in GMM. In this case, the training algorithm may converge to a trivial solution if the covariance matrix is singular. We prevent this singularity problem by adding $1e-6$ to the diagonal entries of the covariance matrices.

### 3.1.2 HYPERPARAMETERS OPTIMIZATION

Once the representation learning pipeline is constructed, we then emphasize on optimizing the hyperparameters for the given pipeline. Here we make use of the Bayesian Optimization (BO) (Shahriari et al., 2015) to tackle this $\boldsymbol{\Theta} - \mathbf{max}$ task, as given below,

$$
\begin{aligned}
\max_{\Theta^C, \Theta^D} & f(\mathrm{K}, \{\Theta^C, \Theta^D\}) + \chi_C(\Theta^C) + \chi_D(\Theta^D), \\
\chi_C(\Theta^C) &= \begin{cases} 0, & \text{if } \Theta^C \in C \\ -\infty, & \text{else} \end{cases}, \quad \chi_D(\Theta^D) = \begin{cases} 0, & \text{if } \Theta^D \in D \\ -\infty, & \text{else} \end{cases},
\end{aligned}
\tag{6}
$$

where the set $C$ and $D$ denote respectively feasible region of the hyperparameters, and $f(\cdot)$ is the objective function given in problem (1). $\chi_C(\Theta^C)$ and $\chi_D(\Theta^D)$ are penalty terms that make sure the hyperparameters fall in the feasible region. Unlike random search (Bergstra & Bengio, 2012) and grid search (Syarif et al., 2016), BO is able to optimize hyperparameters more efficiently. More details about BO are discussed in Appendix A. Algorithm 1 depicts the main steps in TimeAutoML and more details are given in Appendix B.

---

**Algorithm 1:** TimeAutoML with Contrastive Learning

---

**Input:** Maximum iterations $L$ and Bayesian Optimization iterations $B$.

**for** $t = 1, 2, \cdots, L$ **do**

    **Configuring a complete AutoML pipeline** accoding to Thompson Sampling.

    **for** $b = 1, 2, \cdots, B$ **do**

        **Hyperparameters optimization:** Update hyperparameters according to Bayesian Optimization and obtain the objective function value.

    **end**

    **Update** parameters of Thompson Sampling in each module according to the obtained objective function values.

**end**

**Output:** Configured AutoML pipeline and optimized hyperparameters.

---

## 3.2 CONTRASTIVE SELF-SUPERVISED LOSS

According to Zhou et al. (2019); Kieu et al. (2019); Yoon et al. (2019), the structure of the encoder has a direct impact on the representation learning performance. Take anomaly detection as an example, the semi-supervised anomaly detection methods (Pang et al., 2019; Ruff et al., 2019) assume that there are a few labeled anomaly samples in the training dataset, which is more effective in representation learning than unsupervised methods. Instead, the proposed contrastive self-supervised loss does not require any labeled anomaly samples. It uses generated negative samples as anomalies for model building. The goal is to allow the encoder to distinguish the positive samples from generated negative samples.

Given a normal time series $\underline{\boldsymbol{x}}_i \in \mathbb{R}^T$, which is deemed as a positive sample. We then generate the negative sample $\underline{\boldsymbol{x}}_i^{neg}$ by adding noise randomly over a few selected timestamps of $\underline{\boldsymbol{x}}_i$, that is, $\underline{\boldsymbol{x}}_i^{neg} = g(\underline{\boldsymbol{x}}_i)$, where $g(\cdot)$ is the negative sample generation trick. In the experiment, the noise amplitude is randomly selected within the interval $[\min(\underline{\boldsymbol{x}}_i), \max(\underline{\boldsymbol{x}}_i)]$.

In the experiment, we generate one negative sample $\underline{\boldsymbol{x}}_i^{\text{neg}}$ for each positive sample $\underline{\boldsymbol{x}}_i$. The proposed contrastive self-supervised loss $L_{\text{self}}$ aims to distinguish the positive time series sample from the negative ones, which can be given as:

$$
\begin{aligned}
\underline{\boldsymbol{h}}_i^{\text{pos}} &= f_{\text{en}}(\underline{\boldsymbol{x}}_i), \quad \underline{\boldsymbol{h}}_i^{\text{neg}} = f_{\text{en}}(\underline{\boldsymbol{x}}_i^{\text{neg}}), \\
\underline{\boldsymbol{o}}_i^{\text{pos}} &= f_{\text{clas}}(\underline{\boldsymbol{h}}_i^{\text{pos}}), \quad \underline{\boldsymbol{o}}_i^{\text{neg}} = f_{\text{clas}}(\underline{\boldsymbol{h}}_i^{\text{neg}}), \\
L_{\text{self}}(\underline{\boldsymbol{x}}_i) &= f_{\text{BCE}}(\underline{\boldsymbol{o}}_i^{\text{pos}}, 0) + f_{\text{BCE}}(\underline{\boldsymbol{o}}_i^{\text{neg}}, 1),
\end{aligned}
\tag{7}
$$

where $\underline{\boldsymbol{h}}_i^{\text{pos}} \in \mathbb{R}^S$ and $\underline{\boldsymbol{h}}_i^{\text{neg}} \in \mathbb{R}^S$ are respectively the latent space representations of positive samples and negative samples, $S$ represents the length of latent space representation. $f_{\text{clas}}$ is the auxiliary classification network, $\underline{\boldsymbol{o}}_i^{\text{pos}} \in \mathbb{R}^1$ and $\underline{\boldsymbol{o}}_i^{\text{neg}} \in \mathbb{R}^1$ are the outputs of the classifier. $f_{\text{BCE}}$ represents binary cross entropy and we label the positive time series and negative time series as 0 and 1, respectively. More details about the proposed self-supervised loss $L_{\text{self}}$ are shown in Figure 1, we can see that minimizing $L_{\text{self}}$ allows the encoder to distinguish the positive samples from the negative samples in the latent space, and consequently entails better latent space representations.

## 3.3 OVERALL LOSS FUNCTION AND JOINT OPTIMIZATION

Given a dataset with $N$ time series, for fixed pipeline configuration and hyperparameters, the neural network is trained by minimizing an overall loss function containing three parts:

$$
L_{\text{overall}} = \frac{1}{N} \sum_{i=1}^{N} D(\underline{\boldsymbol{x}}_i, \underline{\boldsymbol{x}}_i^{\text{rec}}) + \lambda_1 \frac{1}{N} \sum_{i=1}^{N} E(\underline{\boldsymbol{y}}_i) + \lambda_2 \frac{1}{N} \sum_{i=1}^{N} L_{\text{self}}(\underline{\boldsymbol{x}}_i), \tag{8}
$$

where $D(\underline{\boldsymbol{x}}_i, \underline{\boldsymbol{x}}_i^{\text{rec}})$ represents reconstruction error. $E(\underline{\boldsymbol{y}}_i)$ is the sample energy function which represents the level of abnormality for a given sample. $L_{\text{self}}(\underline{\boldsymbol{x}}_i)$ is the proposed contrastive self-supervised loss. $\lambda_1$ and $\lambda_2$ are two weighting factors governing the trade-off among these three parts.

## 4 EXPERIMENT

The performance of the proposed time series representation learning framework has been assessed via two machine learning tasks, i.e., anomaly detection and clustering. The primary goal is to answer the following two questions in the experiment, 1) **Effectiveness**: can the proposed representation learning framework effectively model and capture the temporal dynamics of a time series? 2) **Robustness**: can TimeAutoML remain effective in the presence of irregularities in sampling rates and contaminated training data?

**Dataset** We first conduct experiments on a total of 85 UCR univariate time series datasets (Chen et al., 2015) to assess the anomaly detection performance. Next, we also assess the performance of the proposed TimeAutoML on a multitude of UEA multivariate time series datasets (Bagnall et al., 2018). We follow the method proposed in Chandola et al. (2008) to create the training, validation, and testing dataset. AUC (Area under the Receiver Operating Curve) is employed to evaluate the anomaly detection performance. For clustering, we carry out experiments on a total of 3 UCR univariate datasets and 2 UEA multivariate datasets. NMI (Normalized Mutual Information) is used to evaluate the clustering results.

**Baselines**   For anomaly detection, the proposed TimeAutoML is compared with a set of state-of-the-art methods including latent ODE (Rubanova et al., 2019), Local Outlier Factor (LoF) (Breunig et al., 2000), Isolation Forest (IF) (Liu et al., 2008), One-Class SVM (OCSVM) (Schölkopf et al., 2001), GRU-AE (Malhotra et al., 2016), DAGMM (Zong et al., 2018) and BeatGAN (Zhou et al., 2019). For clustering, the baseline algorithms for comparison include K-means, GMM, K-means+DTW, K-means+EDR (Chen et al., 2005), K-shape (Paparrizos & Gravano, 2015), SPIRAL (Lei et al., 2017), DEC (Xie et al., 2016), IDEC (Guo et al., 2017), DTC (Madiraju et al., 2018), DTCR (Ma et al., 2019).

## 4.1   ANOMALY DETECTION

We present the AUC scores of the proposed TimeAutoML and other state-of-the-art anomaly detection methods for the 85 univariate time series datasets of UCR archive (Chen et al., 2015). Due to the space limitation, we choose a portion of the time series datasets and the corresponding anomaly detection results are summarized in Table 2. The anomaly detection results for the remaining datasets are summarized in Appendix D, Table A2, A3. It is seen that TimeAutoML achieves best anomaly detection performance over the majority $> 90\%$ of the UCR datasets no matter the time series are regularly or irregularly sampled. In addition, we evaluate the performance of TimeAutoML on a multitude of multivariate time series datasets from UEA archive (Bagnall et al., 2018).

**Effectiveness**   We assess the anomaly detection performance when time series are irregularly sampled ($\beta = 0.5$) and regularly sampled ($\beta = 0$), where $\beta$ is the irregular sampling rate representing the ratio of missing timestamps to all timestamps (Chen et al., 2018b). Note that we put emphasis on analyzing time series with missing timestamps, which is a special type of irregular sampling time series. Table 2 presents the AUC scores of the proposed TimeAutoML and state-of-the-art anomaly detection methods on a selected group of UCR datasets and UEA datasets. We observe that the performance of BeatGAN severely degrades in the presence of irregularities in sampling rates since it is designed for fixed-length input vectors. We also notice that the proposed TimeAutoML exhibits superior performance over existing state-of-the-art anomaly detection methods in almost all cases for irregularly sampled time series. In addition, the negative sampling combined with the contrastive loss function can further boost the anomaly detection performance.

Table 2: AUC scores of TimeAutoML and state-of-the-art anomaly detection methods when time seires are regularly sampled ($\beta = 0$) and irregularly sampled ($\beta = 0.5$). Bold and underlined scores respectively represent the best and second-best performing methods.

| Model | ECG200 | | ECGFiveDays | | GunPoint | | ItalyPD | | MedicalImages | | MoteStrain | | FingerMovements | | LSST | | RacketSports | | PhonemeSpectra | | Heartbeat | |
|---|---|---|---|---|---|---|---|---|---|---|---|---|---|---|---|---|---|---|---|---|---|---|
| | $\beta=0$ | $\beta=0.5$ | $\beta=0$ | $\beta=0.5$ | $\beta=0$ | $\beta=0.5$ | $\beta=0$ | $\beta=0.5$ | $\beta=0$ | $\beta=0.5$ | $\beta=0$ | $\beta=0.5$ | $\beta=0$ | $\beta=0.5$ | $\beta=0$ | $\beta=0.5$ | $\beta=0$ | $\beta=0.5$ | $\beta=0$ | $\beta=0.5$ | $\beta=0$ | $\beta=0.5$ |
| LOF | 0.6271 | 0.6154 | 0.5783 | 0.4856 | 0.5173 | 0.4392 | 0.6061 | 0.5307 | 0.6035 | 0.5398 | 0.5173 | 0.4691 | 0.5489 | 0.5489 | 0.6492 | 0.6492 | 0.4418 | 0.4418 | 0.5646 | 0.5646 | 0.5527 | 0.5527 |
| IF | 0.6953 | 0.6854 | 0.6971 | 0.6653 | 0.4527 | 0.4329 | 0.6358 | 0.5219 | 0.6059 | 0.5181 | 0.6217 | 0.6095 | 0.5796 | 0.5796 | 0.6185 | 0.6185 | 0.5012 | 0.5000 | 0.5355 | 0.5123 | 0.5329 | 0.5329 |
| GRU-ED | 0.7001 | 0.6504 | 0.7412 | 0.5558 | 0.5657 | 0.5247 | 0.8289 | 0.6529 | 0.6619 | 0.5996 | 0.7084 | 0.6149 | 0.5918 | 0.6020 | 0.7412 | 0.6826 | 0.7163 | 0.6511 | 0.5401 | 0.5241 | 0.6189 | 0.6072 |
| DAGMM | 0.5729 | 0.5096 | 0.5732 | 0.5358 | 0.4701 | 0.4701 | 0.7994 | 0.5299 | 0.6473 | 0.5312 | 0.5755 | 0.5474 | 0.5332 | 0.5332 | 0.5113 | 0.4971 | 0.3953 | 0.3953 | 0.5262 | 0.5262 | 0.6048 | 0.5874 |
| BeatGAN | 0.8441 | 0.6932 | 0.9012 | 0.5621 | 0.7587 | 0.6564 | 0.9798 | 0.6214 | 0.6735 | 0.5908 | 0.8201 | 0.7568 | 0.6945 | 0.5304 | 0.7296 | 0.6898 | 0.6289 | 0.5757 | 0.4628 | 0.4393 | 0.6431 | 0.6184 |
| Latent ODE | 0.8214 | 0.8172 | 0.6111 | 0.6037 | 0.8479 | 0.8125 | 0.8221 | 0.7122 | 0.6306 | 0.6292 | 0.7348 | 0.7129 | 0.8017 | 0.7755 | 0.6828 | 0.6636 | 0.9363 | 0.9116 | 0.6813 | 0.6537 | 0.6577 | 0.6468 |
| TimeAutoML without $L_{self}$ | 0.9442 | 0.9012 | 0.9851 | 0.9499 | 0.9307 | 0.9063 | 0.9879 | 0.8481 | 0.7607 | 0.7496 | 0.9207 | 0.8867 | 0.9367 | 0.9204 | 0.7804 | 0.7749 | 0.9825 | 0.9767 | 0.8567 | 0.8459 | 0.7791 | 0.7567 |
| **TimeAutoML** | **0.9712** | **0.9349** | **0.9963** | **0.9519** | **0.9362** | **0.9093** | **0.9959** | **0.8811** | **0.8021** | **0.7693** | **0.9336** | **0.9186** | **0.9745** | **0.9643** | **0.7965** | **0.7827** | **0.9983** | **0.9826** | **0.8817** | **0.8685** | **0.8031** | **0.7703** |
| **Improvement** | **12.47%** | **11.77%** | **9.51%** | **28.66%** | **8.83%** | **9.68%** | **1.61%** | **16.89%** | **12.86%** | **14.01%** | **11.35%** | **16.18%** | **17.28%** | **18.88%** | **5.53%** | **9.29%** | **6.20%** | **7.10%** | **20.04%** | **21.48%** | **14.54%** | **12.35%** |

**Robustness**   We investigate how the proposed TimeAutoML responds to contaminated training data when time series are irregularly sampled with the rate $\beta = 0.5$. AUC scores of the proposed TimeAutoML when training on contaminated data are presented in Appendix D, Table A4, A5. We observe that the anomaly detection performance of TimeAutoML slightly degrades when training data are contaminated. Next, we investigate how TimeAutoML responds to different irregular sampling rate, i.e., when $\beta$ varies from 0 to 0.7. The AUC scores of TimeAutoML and state-of-the-art anomaly detection methods on ECGFiveDays dataset are presented in Fig 2 and the results on other datasets are presented in Appendix D, Fig A1, A2. We notice that TimeAutoML performs well robustly across multiple irregular sampling rates.

## 4.2 VISUALIZATION

In this section, we use a synthetic dataset to elucidate the underlying mechanism of TimeAutoML model for detecting time series anomalies. Figure 3 shows the latent space representation learned via TimeAutoML model from a synthetic dataset. In this dataset, smooth Sine curves are normal time series. The anomaly time series is created by adding noise to the normal time series over a short interval. It is evident from Figure 3 that the latent space representations of normal time series lie in a high density area that can be well characterized by a GMM; while the abnormal time series appears to deviate from the majority of the observations in the latent space. In short, the proposed encoder-decoder structure allows us to project the time series data in the original space onto vector representations in the latent space. In doing so, we can detect anomalies via clustering-based methods, e.g., GMM, and easily visualize as well as interpret the detected time series anomalies.

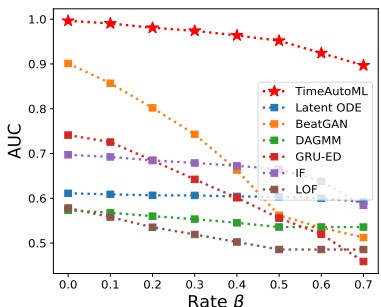

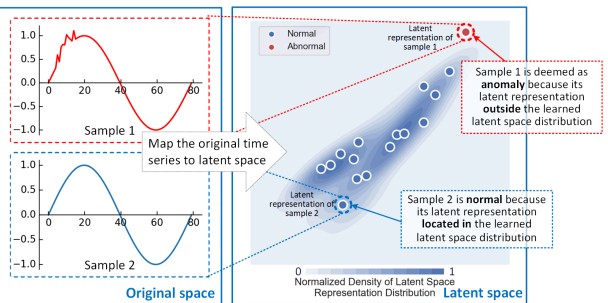

Figure 2: AUC scores of TimeAutoML on ECGFiveDays dataset when irregular sampling rate $\beta$ varies from 0 to 0.7.

Figure 3: Anomaly interpretation via analysis in latent space.

## 4.3 CLUSTERING

Apart from anomaly detection, TimeAutoML can be tailored for other machine learning tasks as well, e.g., multi-class clustering. In particular, the clustering process is carried out in the latent space via the GMM model, along with other modules in the pipeline.

We evaluate the effectiveness of TimeAutoML on three univariate time series datasets as well as two multivariate time series datasets. The NMI of TimeAutoML and state-of-the-art clustering methods are shown in Table 3. We observe that TimeAutoML generally achieves superior performance compared to baseline algorithms. This is because: i) it can automatically select the best module and hyperparameters; ii) the auxiliary classification task can enhance its representation capability.

Table 3: NMI scores of TimeAutoML and state-of-the-art clustering methods. Bold and underlined scores respectively represent the best and second-best performing methods.

| Model | GunPoint | | ECGFiveDays | | ProximalPOAG | | AtrialFibrillation | | Epilepsy | |
|---|---|---|---|---|---|---|---|---|---|---|
| | $\beta = 0$ | $\beta = 0.5$ | $\beta = 0$ | $\beta = 0.5$ | $\beta = 0$ | $\beta = 0.5$ | $\beta = 0$ | $\beta = 0.5$ | $\beta = 0$ | $\beta = 0.5$ |
| K-means | 0.0011 | 0.0185 | 0.0002 | 0.0020 | 0.4842 | 0.0076 | 0 | 0 | 0.0760 | 0.1370 |
| GMM | 0.0063 | 0.0090 | 0.0030 | 0.0019 | 0.5298 | 0.0164 | 0 | 0 | 0.1276 | 0.0828 |
| K-means+DTW | 0.2100 | 0.0766 | 0.2508 | 0.0081 | 0.4830 | 0.4318 | 0.0650 | 0.1486 | 0.1454 | 0.1534 |
| K-means+EDR | 0.0656 | 0.0692 | 0.1614 | 0.0682 | 0.1105 | 0.0260 | 0.2025 | 0.1670 | 0.3064 | 0.2934 |
| K-shape | 0.0011 | 0.0280 | 0.7458 | 0.0855 | 0.4844 | 0.0237 | 0.3492 | 0.2841 | 0.2339 | 0.1732 |
| SPIRAL | 0.0020 | 0.0019 | 0.0218 | 0.0080 | 0.5457 | 0.0143 | 0.2249 | 0.1475 | 0.1600 | 0.1912 |
| DEC | 0.0263 | 0.0261 | 0.0148 | 0.1155 | 0.5504 | 0.1415 | 0.1242 | 0.1084 | 0.2206 | 0.1971 |
| IDEC | 0.0716 | 0.0640 | 0.0548 | 0.1061 | 0.5452 | 0.1122 | 0.1132 | 0.1242 | 0.2295 | 0.2372 |
| DTC | 0.3284 | 0.0714 | 0.0170 | 0.0162 | 0.4154 | 0.0263 | 0.1443 | 0.1331 | 0.2036 | 0.0886 |
| DTCR | 0.0564 | 0.0676 | 0.3299 | 0.1415 | 0.5190 | 0.3392 | 0.4081 | 0.3593 | 0.3827 | 0.2583 |
| TimeAutoML without $L_{self}$ | 0.3262 | 0.2794 | 0.5914 | 0.3220 | 0.5915 | 0.5051 | 0.6623 | 0.6469 | 0.5073 | 0.4735 |
| **TimeAutoML** | **0.3323** | **0.2841** | 0.6108 | **0.3476** | **0.5981** | **0.5170** | **0.6871** | **0.6649** | **0.5419** | **0.5056** |

## 5 CONCLUSION

Representation learning on irregularly sampled time series is an under-explored topic. In this paper we propose a TimeAutoML framework to carry out unsupervised autonomous representation learning for irregularly sampled multivariate time series. In addition, we propose a self-supervised loss function to get labels directly from the unlabeled data. Strong empirical performance has been observed for TimeAutoML on a plurality of real-world datasets. While tremendous efforts have been undertaken for time series learning in general, AutoML for time series representation learning is still in its infancy and we hope the findings in this paper will open up new venues along this direction and spur further research efforts.

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

# A APPENDIX A: BAYESIAN OPTIMIZATION

Let $f(\theta)$ denote the objective function. Given function values during the preceding $T$ iterations $\underline{v} = [f(\theta_1), f(\theta_2), \cdots, f(\theta_T)]$, we pick up the variable for sampling in the next iteration via solving the maxmization problem that involves the acquisition function i.e., expected improvement (EI) based on the postetior GP model.

Specifically, the objective function is assumed to follow a GP model (Shahriari et al., 2015) and can be expressed as $f(\theta) \sim GP(m(\theta), \boldsymbol{K})$, where $m(\theta)$ represents the mean function. And $\boldsymbol{K}$ represents the covariance matrix of $\{\theta_i\}_{i=0}^{T}$, namely, $\boldsymbol{K}_{ij} = \kappa(\theta_i, \theta_j)$, where $\kappa(\cdot, \cdot)$ is the kernel function. In particular, the poster probability of $f(\theta)$ at iteration $T + 1$ is assumed to follow a Gaussian distribution with mean $\mu(\theta^*)$ and covariance $\sigma^2(\theta^*)$, given the observation function values $\underline{v}$ :

$$
\begin{aligned}
\mu(\theta^*) &= \underline{\boldsymbol{\kappa}}^T[\boldsymbol{K} + \sigma_n^2\mathrm{I}]^{-1}\underline{v}, \\
\sigma^2(\theta^*) &= \kappa(\theta^*, \theta^*) - \underline{\boldsymbol{\kappa}}^T[\boldsymbol{K} + \sigma_n^2\mathrm{I}]^{-1}\underline{\boldsymbol{\kappa}},
\end{aligned}
\tag{9}
$$

where $\underline{\boldsymbol{\kappa}}$ is a vector of covariance terms between $\theta^*$ and $\{\theta_i\}_{i=0}^{T}$, and $\sigma_n^2$ denotes the noise variance. We choose the kenel function as ARD Matérn 5/2 kernel (Shahriari et al., 2015) in this paper:

$$
\kappa(\underline{\boldsymbol{p}}, \underline{\boldsymbol{p}}') = \tau_0^2 \exp(-\sqrt{5}r)(1 + \sqrt{5}r + \frac{5}{3}r^2),
\tag{10}
$$

where $\underline{\boldsymbol{p}}$ and $\underline{\boldsymbol{p}}'$ are input vectors, $r^2 = \sum_{i=1}^{d} (\underline{\boldsymbol{p}}_i - \underline{\boldsymbol{p}}_i')^2/\tau_i^2$ , and $\psi = \{\{\tau_i\}_{i=0}^{d}, \sigma_n^2\}$ are the GP hyperparameters which are determined by minimizing the negative log marginal likelihood $\log(y|\psi)$ :

$$
\min_{\psi} \log \det(\boldsymbol{K} + \sigma_n^2\mathrm{I}) + \underline{v}^T(\boldsymbol{K} + \sigma_n^2\mathrm{I})^{-1}\underline{v}.
\tag{11}
$$

Given the mean $\mu(\theta^*)$ and covariance $\sigma^2(\theta^*)$ in (9), $\theta_{T+1}$ can be obtained via solving the following optimization problem:

$$
\begin{aligned}
\theta_{T+1} &= \arg\max_{\theta^*} \mathrm{EI}(\theta^*) \\
&= \arg\max_{\theta^*} (\mu(\theta^*) - y^+)\Phi(\frac{\mu(\theta^*) - y^+}{\sigma(\theta^*)}) + \sigma\phi(\frac{\mu(\theta^*) - y^+}{\sigma(\theta^*)}),
\end{aligned}
\tag{12}
$$

where $y^+ = \max[f(\theta_1), f(\theta_2), \cdots, f(\theta_T)]$ represents the maximum observation value in the previous $T$ iterations. $\Phi$ is normal cumulative distribution function and $\phi$ is normal probability density function. Through maximizing the EI acquisition function, we seek to improve $f(\theta_{T+1})$ monotonically after each iteration.

## B   APPENDIX B: DETAILED VERSION OF TIMEAUTOML

---

**Algorithm 2:** Detailed Version of TimeAutoML

---

**Input:** $L$: pre-defined threshold for maximum number of iterations. $B$: pre-defined threshold for maximum Bayesian optimization iterations. $\underline{\boldsymbol{\alpha}}_0$ and $\underline{\boldsymbol{\beta}}_0$: pre-defined Beta distribution priors. $f_{\text{upp}}$ and $f_{\text{low}}$: pre-defined upper bound and lower bound to objective function $f$.

**Set:** $\underline{\boldsymbol{\alpha}}_i(t) \in \mathbb{R}^{Q_i}, \underline{\boldsymbol{\beta}}_i(t) \in \mathbb{R}^{Q_i}$: the cumulative reward and punishment for $i^{th}$ module for the $t^{th}$ iteration, specifically, $\underline{\boldsymbol{\alpha}}_i(1) = \underline{\boldsymbol{\alpha}}_0$ and $\underline{\boldsymbol{\beta}}_i(1) = \underline{\boldsymbol{\beta}}_0$.

**for** $t = 1, 2, \cdots, L$ **do**

   **for** $i = 1, 2, \cdots, M$ **do**

     | Sample $\underline{\boldsymbol{w}}_i \sim \text{Beta}(\underline{\boldsymbol{\alpha}}_i(t), \underline{\boldsymbol{\beta}}_i(t))$

   **end**

   **Obtain the TimeAutoML pipeline configuration by solving the following optimization problem:**

$$\underset{\text{K}}{\text{maxmize}} \sum_{i=1}^{M} (\underline{\boldsymbol{k}}_i)^{\top} \underline{\boldsymbol{w}}_i \quad \text{subject to } \underline{\boldsymbol{k}}_i \in \{0,1\}^{Q_i}, 1^{\top}\underline{\boldsymbol{k}}_i = 1, \forall i \in [M],$$

   **for** $b = 1, 2, \cdots, B$ **do**

     | **Hyperparameters optimization:** Update hyperparameters according to Bayesian optimization framework given in Appendix A and the obtained objective function value is denoted as $f(\text{K}(t), \Theta_b(t))$, where $\text{K}(t)$ denote the module options in $t^{th}$ iteration and $\Theta_b(t)$ denote the hyperparameters in $b^{th}$ Bayesian optimization iteration in $t^{th}$ iteration.

   **end**

   **Update Beta distribution of the options in the configured pipeline:**

   1. Let $f(t) = \max\{f(\text{K}(t), \Theta_b(t)), b = 1, \cdots, B\}$ denote the performance of TimeAutoML model at the $t^{th}$ iteration.

   2. Compute the continuous reward $\widetilde{r}$:

$$\widetilde{r} = \max\{0, \tfrac{f(t) - f_{\text{low}}}{f_{\text{upp}} - f_{\text{low}}}\}$$

   3. Obtain the binary reward $r \sim \text{Bernoulli}(\widetilde{r})$.

   **for** $i = 1, 2, \cdots, M$ **do**

$$\underline{\boldsymbol{\alpha}}_i(t+1) = \underline{\boldsymbol{\alpha}}_i(t) + \underline{\boldsymbol{k}}_i \cdot r$$
$$\underline{\boldsymbol{\beta}}_i(t+1) = \underline{\boldsymbol{\beta}}_i(t) + \underline{\boldsymbol{k}}_i \cdot (1 - r)$$

   **end**

**end**

**Output:** A TimeAutoML model with optimized hyperparameters.

---

It is seen that TimeAutoML consists of two main stages, i.e., pipeline configuration and hyperparameter optimization. In every iteration of TimeAutoML, Thompson sampling is utilized to refine the pipeline configuration at first. After that, Bayesian optimization is invoked to optimize the hyperparameters of the model. Finally, the Beta distribution of the chosen options will be updated according to the performance of the configured pipeline.

In the experiment, the upper limit to number of entire TimeAutoML iterations, BO iterations are set as 40 and 25 respectively. The Beta distribution priors are set respectively as $\alpha_0 = 10$ and $\beta_0 = 10$.

# C APPENDIX C: SEARCH SPACE: OPTIONS AND HYPERPARAMETERS

Table A1: Modules, options, and hyperparameters of TimeAutoML.

| Module | Options | Hyperparameters |
|---|---|---|
| Data augmentation | Scaling | Continuous and discrete hyperparameters |
| | Shifting | Discrete hyperparameters |
| | Time-warping | Discrete hyperparameters |
| Encoder | RNN | Discrete hyperparameters |
| | LSTM | Discrete hyperparameters |
| | GRU | Discrete hyperparameters |
| Attention | No attention | None |
| | Self-attention | None |
| Decoder | RNN | Discrete hyperparameters |
| | LSTM | Discrete hyperparameters |
| | GRU | Discrete hyperparameters |
| EM Estimator | Gaussian Mixture Model | Discrete hyperparameters |
| Similarity Selection | Relative Euclidean distance | None |
| | Cosine similarity | None |
| | Both | None |
| Estimation Network | Multi-layer feed-forward neural network | Discrete hyperparameters |
| Auxiliary Classification Network | Multi-layer feed-forward neural network | Discrete hyperparameters |

## C.1 DATA AUGMENTATION

- **Scaling:** increasing or decreasing the amplitude of the time series. There are two hyperparametes, the number of data augmentation samples $N^{\mathrm{aug}} \in [0, 100]$ and the scaling size $h^{\mathrm{amp}} \in [0.5, 1.8]$.

- **Shifting:** cyclically shifting the time series to the left or right. There are two hyperparametes, the number of data augmentation samples $N^{\mathrm{aug}} \in [0, 100]$ and the shift size $h^{\mathrm{shift}} \in [-10, 10]$.

- **Time-warping:** randomly "slowing down" some timestamps and "speeding up" some timestamps. For each timestamp to "speed up", we delete the data value at that timestamp. For each timestamp to "slow down", we insert a new data value just before that timestamp. There are two hyperparametes, the number of data augmentation samples $N^{\mathrm{aug}} \in [0, 100]$ and the number of time-warping timestamps $h^{\mathrm{tm}} \in [T/10, T/4]$.

## C.2 ENCODER

For all encoders, i.e. RNN, LSTM, and GRU, there is only one hyperparameter, i.e., the size of the encoder hidden state. And we assume it is no larger than 32, i.e., $h^{\mathrm{enc}} \in [1, 32]$.

## C.3 ATTENTION

Self-attention mechanism has been considered in this framework.

## C.4 DECODER

For all decoders, i.e. RNN, LSTM, and GRU, there is only one hyperparameter, i.e., the size of decoder hidden state $h^{\mathrm{dec}}$. For univariate time series, we assume it is no larger than 32, i.e., $h^{\mathrm{dec}} \in [1, 32]$. And we assume $h^{\mathrm{dec}} \in [n^{\mathrm{feat}}, 4 * n^{\mathrm{feat}}]$ for multivariate time series, where $n^{\mathrm{feat}}$ represents the dimension of the multivariate time series.

## C.5 EM ESTIMATOR

In this module, we provide a statistical model GMM to carry out latent space representation distribution estimation. There is one hyperparameter, the number of mixture-component of GMM. EM algorithm is used to estimate the key parameters of GMM.

## C.6 SIMILARITY SELECTION

We offer three similarity functions for selection, including relative Euclidean distance, cosine similarity, or the concatenation of both.

## C.7 ESTIMATION NETWORK

We utilize a multi-layer neural network as the estimation network in our pipeline. There are two hyperparameters to be optimized, i.e., the number of layers $e^{\mathrm{layer}} \in [1, 5]$ and the number of nodes in each layer $e_i^{\mathrm{node}} \in [8, 128]$, $\forall 1 \leq i \leq e^{\mathrm{layer}}$.

## C.8 AUXILIARY CLASSIFICATION NETWORK

We utilize a multi-layer neural network as the auxiliary classification network in our pipeline. There are two hyperparameters to be optimized, i.e., the number of layers $c^{\mathrm{layer}} \in [1, 5]$ and the number of nodes in each layer $c_i^{\mathrm{node}} \in [8, 128]$, $\forall 1 \leq i \leq c^{\mathrm{layer}}$.

# D  APPENDIX D: RESULT

## D.1  ANOMALY DETECTION PERFORMANCE FOR UNIVARIATE TIME SERIES

Table A2: AUC scores of TimeAutoML and state-of-the-art anomaly detection methods on UCR time series dataset when time series are regularly sampled ($\beta = 0$). Bold and underlined scores respectively represent the best and second-best performing methods.

| dataset | TimeAutoML | Latent ODE | BeatGAN | DAGMM | GRU-ED | IF | LOF |
|---|---|---|---|---|---|---|---|
| Adiac | 1 | 1 | 1 | 1 | 1 | 0.9375 | 0.4375 |
| ArrowHead | **0.9876** | 0.8592 | 0.7923 | 0.872 | 0.4008 | 0.7899 | 0.442 |
| Beef | 1 | 1 | 1 | 1 | 0.8333 | 1 | 0.4167 |
| BeetleFly | 1 | 1 | 1 | 1 | 1 | 0.35 | 0.4 |
| BirdChicken | 1 | 1 | 0.8 | 0.9 | 0.6 | 0.5 | 0.4 |
| Car | 1 | 1 | 0.6233 | 0.3346 | 1 | 0.2854 | 0.4231 |
| CBF | 1 | 0.6573 | 0.9909 | 0.7983 | 0.8606 | 0.6408 | 0.9399 |
| ChlorineConcentration | **0.6653** | 0.5672 | 0.5291 | 0.5724 | 0.5048 | 0.5449 | 0.5899 |
| CinCECGTorso | 0.8951 | 0.6761 | **0.9966** | 0.7908 | 0.4958 | 0.6749 | 0.9641 |
| Coffee | 1 | 1 | 1 | 1 | 0.9333 | 0.75 | 0.7167 |
| Computers | **0.8354** | 0.744 | 0.738 | 0.6563 | 0.7686 | 0.468 | 0.5714 |
| CricketX | 1 | 0.9744 | 0.8754 | 0.8123 | 0.7892 | 0.7405 | 0.6282 |
| CricketY | 1 | 0.954 | 0.9828 | 0.8997 | 0.931 | 0.8161 | 0.9827 |
| CricketZ | 1 | 0.9583 | 0.8285 | 0.6897 | 0.8333 | 0.6521 | 0.6249 |
| DiatomSizeReduction | 1 | 0.8571 | 1 | 1 | 0.9913 | 0.9783 | 0.9946 |
| DistalPhalanxOutlineAgeGroup | **0.9912** | 0.8333 | 0.8 | 0.8333 | 0.6879 | 0.7021 | 0.6858 |
| DistalPhalanxOutlineCorrect | **0.8626** | 0.8333 | 0.5342 | 0.6721 | 0.6193 | 0.6204 | 0.7693 |
| DistalPhalanxTW | 1 | 0.9143 | 1 | 1 | 1 | 0.9643 | 1 |
| Earthquakes | **0.8418** | 0.7421 | 0.6221 | 0.5529 | 0.8033 | 0.5671 | 0.5428 |
| ECG5000 | **0.9981** | 0.5648 | 0.9923 | 0.8475 | 0.8998 | 0.9304 | 0.5436 |
| ElectricDevices | **0.8427** | 0.5626 | 0.8381 | 0.7172 | 0.7958 | 0.5518 | 0.5528 |
| FaceAll | 1 | 0.7674 | 0.9821 | 0.9841 | 0.9844 | 0.7639 | 0.7847 |
| FaceFour | 1 | 1 | 1 | 1 | 0.9286 | 0.9286 | 0.4286 |
| FacesUCR | 1 | 0.6368 | 0.9276 | 0.9065 | 0.8786 | 0.6782 | 0.8296 |
| FiftyWords | 1 | 0.8187 | 0.9895 | 0.9901 | 0.5643 | 0.9474 | 0.807 |
| Fish | 1 | 0.9394 | 0.8523 | 0.7273 | 0.5909 | 0.4772 | 0.6212 |
| FordA | 0.6229 | 0.6204 | 0.5496 | 0.5619 | **0.6306** | 0.4963 | 0.4708 |
| FordB | 0.6008 | **0.6212** | 0.5999 | 0.6021 | 0.5949 | 0.5949 | 0.4971 |
| Ham | **0.8961** | 0.8579 | 0.6556 | 0.7667 | 0.6358 | 0.6348 | 0.6296 |
| HandOutlines | 0.8808 | 0.8362 | **0.9031** | 0.8524 | 0.5679 | 0.7349 | 0.7413 |
| Haptics | **0.8817** | 0.8579 | 0.7266 | 0.6698 | 0.5826 | 0.6674 | 0.5167 |
| Herring | 1 | 0.9581 | 0.8333 | 0.6528 | 0.8026 | 0.7231 | 0.7105 |
| InlineSkate | **0.8556** | 0.8039 | 0.65 | 0.7147 | 0.5559 | 0.4223 | 0.6254 |
| InsectWingbeatSound | 0.91 | 0.6574 | 0.9605 | **0.9735** | 0.7549 | 0.7861 | 0.9333 |
| LargeKitchenAppliances | **0.8708** | 0.7703 | 0.5887 | 0.5824 | 0.7975 | 0.5025 | 0.5289 |
| Lightning2 | 1 | 0.9242 | 0.6061 | 0.7574 | 0.5758 | 0.909 | 0.7197 |
| Lightning7 | 1 | 1 | 1 | 1 | 1 | 1 | 0.4211 |
| Mallat | **0.9996** | 0.6639 | 0.9979 | 0.9701 | 0.5728 | 0.8377 | 0.8811 |
| Meat | 1 | 1 | 1 | 0.975 | 1 | 0.7001 | 0.7001 |
| MiddlePhalanxOutlineAgeGroup | 1 | 0.954 | 0.9673 | 0.8512 | 0.7931 | 0.7414 | 0.431 |
| MiddlePhalanxOutlineCorrect | **0.9242** | 0.7355 | 0.4401 | 0.7012 | 0.7013 | 0.4818 | 0.5725 |
| MiddlePhalanxTW | 1 | 0.9524 | 1 | 1 | 1 | 0.9762 | 1 |
| NonInvasiveFetalECGThorax1 | 1 | 0.9167 | 1 | 1 | 1 | 0.9306 | 0.8611 |
| NonInvasiveFetalECGThorax2 | 1 | 0.9028 | 0.9167 | 1 | 1 | 0.9722 | 1 |
| OliveOil | 1 | 1 | 0.9167 | 0.9167 | 0.9167 | 0.9583 | 1 |
| OSULeaf | 1 | 0.8864 | 0.8125 | 0.8892 | 0.8352 | 0.375 | 0.6823 |
| PhalangesOutlinesCorrect | **0.7423** | 0.7049 | 0.4321 | 0.5521 | 0.6625 | 0.5192 | 0.6629 |
| Phoneme | **0.9148** | 0.6823 | 0.7054 | 0.5826 | 0.7964 | 0.4904 | 0.5943 |
| Plane | 1 | 1 | 1 | 1 | 1 | 1 | 0.4 |
| ProximalPhalanxOutlineAgeGroup | **0.998** | 0.8024 | 0.975 | 0.9723 | 0.9614 | 0.82 | 0.775 |
| ProximalPhalanxOutlineCorrect | **0.9255** | 0.6482 | 0.5823 | 0.7221 | 0.9051 | 0.5348 | 0.7474 |
| ProximalPhalanxTW | 1 | 0.8664 | 0.9663 | 0.9623 | 0.9079 | 0.8889 | 0.9311 |
| RefrigerationDevices | **0.9323** | 0.7483 | 0.7264 | 0.5722 | 0.5434 | 0.4665 | 0.5714 |
| ScreenType | **0.8572** | 0.7453 | 0.7453 | 0.5472 | 0.7686 | 0.4921 | 0.5289 |
| ShapeletSim | 1 | 0.9 | 0.7421 | 0.5721 | 0.9728 | 0.5611 | 0.5481 |
| ShapesAll | 1 | 1 | 0.9 | 0.95 | 1 | 0.85 | 0.95 |
| SmallKitchenAppliances | 0.9586 | 0.7151 | 0.6541 | 0.7321 | **0.9621** | 0.6812 | 0.6563 |
| SonyAIBORobotSurface1 | **0.9998** | 0.6886 | 0.9982 | 0.9834 | 0.9991 | 0.8129 | 0.9731 |
| SonyAIBORobotSurface2 | **0.9907** | 0.6211 | 0.9241 | 0.8994 | 0.9236 | 0.5981 | 0.7152 |
| StarLightCurves | **0.9135** | 0.5548 | 0.8083 | 0.8924 | 0.8386 | 0.8161 | 0.5028 |
| Strawberry | 0.7805 | 0.6786 | 0.5659 | 0.5659 | **0.8184** | 0.4738 | 0.4433 |
| SwedishLeaf | **0.9913** | 0.9394 | 0.6963 | 0.5758 | 0.6566 | 0.6212 | 0.6212 |
| Symbols | **0.9987** | 0.7669 | 0.9881 | 0.9762 | 0.947 | 0.8025 | 0.9942 |
| SyntheticControl | 1 | 1 | 0.736 | 0.6524 | 1 | 0.3299 | 0.66 |
| ToeSegmentation1 | **0.9437** | 0.7112 | 0.8819 | 0.6264 | 0.5726 | 0.5226 | 0.6708 |
| ToeSegmentation2 | **0.9907** | 0.8225 | 0.9358 | 0.8243 | 0.6157 | 0.5612 | 0.7021 |
| Trace | 1 | 1 | 1 | 1 | 1 | 0.9211 | 0.4211 |
| TwoLeadECG | **0.9959** | 0.6485 | 0.8759 | 0.6941 | 0.8641 | 0.5967 | 0.8274 |
| TwoPatterns | **0.9996** | 0.5899 | 0.9936 | 0.7163 | 0.9297 | 0.5411 | 0.7371 |
| UWaveGestureLibraryAll | **0.9941** | 0.6487 | 0.9935 | 0.9898 | 0.8106 | 0.9342 | 0.7896 |
| UWaveGestureLibraryX | **0.7477** | 0.6136 | 0.6563 | 0.6796 | 0.6009 | 0.5626 | 0.4696 |
| UWaveGestureLibraryY | **0.9845** | 0.6256 | 0.9742 | 0.9626 | 0.9357 | 0.9159 | 0.6244 |
| UWaveGestureLibraryZ | **0.9957** | 0.6587 | 0.9897 | 0.9883 | 0.9662 | 0.9161 | 0.8671 |
| Wafer | **0.9903** | 0.4947 | 0.9315 | 0.9586 | 0.6763 | 0.9436 | 0.5599 |
| Wine | 1 | 0.9536 | 0.8704 | 0.9074 | 0.7531 | 0.4259 | 0.6689 |
| WordSynonyms | **0.9929** | 0.7862 | 0.9862 | 0.9621 | 0.8245 | 0.8226 | 0.8442 |
| Worms | **0.9968** | 0.8485 | 0.8978 | 0.7677 | 0.7126 | 0.5341 | 0.5896 |
| WormsTwoClass | **0.9583** | 0.9375 | 0.6307 | 0.6957 | 0.7591 | 0.4021 | 0.4432 |
| Yoga | **0.7538** | 0.5823 | 0.6883 | 0.6766 | 0.5884 | 0.5421 | 0.6267 |

Table A3: AUC scores of TimeAutoML and state-of-the-art anomaly detection methods on UCR time series dataset when time series are irregularly sampled ($\beta = 0.5$). Bold and underlined scores respectively represent the best and second-best performing methods.

| dataset | TimeAutoML | Latent ODE | BeatGAN | DAGMM | GRU-ED | IF | LOF |
|---|---|---|---|---|---|---|---|
| Adiac | 1 | 1 | 0.25 | 0.9375 | 1 | 0.4375 | 0.4 |
| ArrowHead | 0.9816 | 0.8095 | 0.7633 | 0.8671 | 0.3478 | 0.7547 | 0.442 |
| Beef | 1 | 1 | 1 | 1 | 1 | 0.5 | 0.4167 |
| BeetleFly | 1 | 1 | 0.9 | 0.75 | 1 | 0.35 | 0.35 |
| BirdChicken | 1 | 1 | 1 | 0.9 | 0.6 | 0.4 | 0.4 |
| Car | 1 | 1 | 0.6154 | 0.3077 | 0.6538 | 0.2692 | 0.4231 |
| CBF | 0.9933 | 0.6362 | 0.8725 | 0.7819 | 0.7638 | 0.5281 | 0.8849 |
| ChlorineConcentration | 0.5954 | 0.5669 | 0.4916 | 0.572 | 0.493 | 0.5171 | 0.5121 |
| CinCECGTorso | 0.876 | 0.6679 | 0.9037 | 0.7855 | 0.4931 | 0.6584 | 0.7881 |
| Coffee | 1 | 1 | 1 | 1 | 0.9333 | 0.75 | 0.4333 |
| Computers | 0.9188 | 0.5723 | 0.6613 | 0.635 | 0.72 | 0.456 | 0.5714 |
| CricketX | 1 | 0.9743 | 0.6731 | 0.8077 | 0.7051 | 0.7372 | 0.6282 |
| CricketY | 0.9897 | 0.9081 | 0.9639 | 0.8966 | 0.6897 | 0.8161 | 0.7989 |
| CricketZ | 1 | 0.9166 | 0.8125 | 0.6875 | 0.8333 | 0.6458 | 0.4375 |
| DiatomSizeReduction | 0.9793 | 0.8467 | 0.8104 | 1 | 0.6772 | 0.4848 | 0.6065 |
| DistalPhalanxOutlineAgeGroup | 0.9516 | 0.8395 | 0.7821 | 0.8333 | 0.6835 | 0.678 | 0.6755 |
| DistalPhalanxOutlineCorrect | 0.764 | 0.6759 | 0.5167 | 0.4941 | 0.5367 | 0.4719 | 0.4766 |
| DistalPhalanxTW | 1 | 0.9 | 1 | 1 | 1 | 0.9107 | 0.6643 |
| Earthquakes | 0.8191 | 0.6608 | 0.6083 | 0.5214 | 0.5841 | 0.5380 | 0.5248 |
| ECG5000 | 0.9765 | 0.5422 | 0.8817 | 0.6795 | 0.8537 | 0.7008 | 0.5142 |
| ElectricDevices | 0.7481 | 0.5621 | 0.5288 | 0.6974 | 0.6679 | 0.5363 | 0.5331 |
| FaceAll | 1 | 0.6545 | 0.8552 | 0.6671 | 0.6892 | 0.6944 | 0.6528 |
| FaceFour | 1 | 1 | 1 | 0.9643 | 0.6429 | 0.6429 | 0.4286 |
| FacesUCR | 0.9491 | 0.6474 | 0.6296 | 0.891 | 0.5507 | 0.6765 | 0.6276 |
| FiftyWords | 0.997 | 0.7456 | 0.9684 | 0.9895 | 0.4532 | 0.8728 | 0.7149 |
| Fish | 0.9697 | 0.9393 | 0.8409 | 0.7727 | 0.4545 | 0.5 | 0.6212 |
| FordA | 0.6157 | 0.6037 | 0.5127 | 0.5414 | 0.6005 | 0.4958 | 0.4684 |
| FordB | 0.5808 | 0.6164 | 0.5212 | 0.587 | 0.5489 | 0.5352 | 0.4971 |
| Ham | 0.8812 | 0.8519 | 0.4778 | 0.7556 | 0.5278 | 0.6296 | 0.6296 |
| HandOutlines | 0.8504 | 0.6442 | 0.7287 | 0.8409 | 0.5425 | 0.6142 | 0.6266 |
| Haptics | 0.9353 | 0.8415 | 0.5547 | 0.6641 | 0.5223 | 0.5882 | 0.5167 |
| Herring | 0.9642 | 0.9474 | 0.7895 | 0.6491 | 0.5855 | 0.7105 | 0.7105 |
| InlineSkate | 0.8928 | 0.7281 | 0.5618 | 0.5691 | 0.5477 | 0.4088 | 0.5026 |
| InsectWingbeatSound | 0.8669 | 0.6515 | 0.8364 | 0.9137 | 0.6444 | 0.7139 | 0.8111 |
| LargeKitchenAppliances | 0.8375 | 0.7686 | 0.5433 | 0.569 | 0.5785 | 0.4905 | 0.4865 |
| Lightning2 | 1 | 0.9015 | 0.4141 | 0.7475 | 0.4545 | 0.909 | 0.4934 |
| Lightning7 | 1 | 1 | 1 | 1 | 1 | 0.9474 | 0.4211 |
| Mallat | 0.9888 | 0.6546 | 0.8948 | 0.9687 | 0.5512 | 0.6375 | 0.881 |
| Meat | 1 | 1 | 0.675 | 0.975 | 0.95 | 0.7001 | 0.7001 |
| MiddlePhalanxOutlineAgeGroup | 1 | 0.9081 | 0.9655 | 0.8448 | 0.6552 | 0.5574 | 0.431 |
| MiddlePhalanxOutlineCorrect | 0.7573 | 0.7195 | 0.4352 | 0.6543 | 0.4344 | 0.4738 | 0.4752 |
| MiddlePhalanxTW | 1 | 0.875 | 1 | 1 | 0.9952 | 0.8048 | 0.5524 |
| NonInvasiveFetalECGThorax1 | 1 | 0.9028 | 0.8889 | 1 | 1 | 0.9583 | 0.7222 |
| NonInvasiveFetalECGThorax2 | 1 | 0.8333 | 0.8148 | 1 | 1 | 0.9028 | 0.7222 |
| OliveOil | 1 | 1 | 0.5883 | 0.9583 | 0.5 | 0.4583 | 0.4167 |
| OSULeaf | 0.9955 | 0.8318 | 0.7184 | 0.7557 | 0.8227 | 0.375 | 0.6659 |
| PhalangesOutlinesCorrect | 0.6819 | 0.6671 | 0.4203 | 0.5372 | 0.5576 | 0.4466 | 0.4864 |
| Phoneme | 0.8898 | 0.6676 | 0.6135 | 0.5631 | 0.7821 | 0.4864 | 0.5692 |
| Plane | 1 | 1 | 1 | 1 | 1 | 1 | 0.4 |
| ProximalPhalanxOutlineAgeGroup | 0.998 | 0.723 | 0.861 | 0.965 | 0.925 | 0.72 | 0.5 |
| ProximalPhalanxOutlineCorrect | 0.8299 | 0.6398 | 0.5625 | 0.7033 | 0.5716 | 0.4972 | 0.4997 |
| ProximalPhalanxTW | 1 | 0.8333 | 0.8948 | 0.9603 | 0.842 | 0.875 | 0.5833 |
| RefrigerationDevices | 0.8799 | 0.6738 | 0.7047 | 0.5597 | 0.5268 | 0.4625 | 0.4284 |
| ScreenType | 0.8446 | 0.6954 | 0.7153 | 0.6137 | 0.8012 | 0.492 | 0.5714 |
| ShapeletSim | 0.9278 | 0.6901 | 0.7358 | 0.5056 | 0.6531 | 0.4444 | 0.5056 |
| ShapesAll | 1 | 1 | 0.9 | 0.95 | 0.9 | 0.85 | 0.95 |
| SmallKitchenAppliances | 0.9538 | 0.6812 | 0.6223 | 0.7218 | 0.9117 | 0.6692 | 0.6563 |
| SonyAIBORobotSurface1 | 0.99 | 0.6833 | 0.8132 | 0.8001 | 0.9605 | 0.5227 | 0.7402 |
| SonyAIBORobotSurface2 | 0.8206 | 0.6144 | 0.6641 | 0.7948 | 0.6927 | 0.5556 | 0.6126 |
| StarLightCurves | 0.9118 | 0.5489 | 0.795 | 0.8874 | 0.8324 | 0.8186 | 0.5028 |
| Strawberry | 0.7427 | 0.6733 | 0.5986 | 0.546 | 0.5672 | 0.4643 | 0.4433 |
| SwedishLeaf | 0.9889 | 0.9318 | 0.6566 | 0.5758 | 0.4949 | 0.4949 | 0.4734 |
| Symbols | 0.9961 | 0.7437 | 0.982 | 0.9521 | 0.9303 | 0.7998 | 0.9636 |
| SyntheticControl | 1 | 0.968 | 0.704 | 0.62 | 0.5836 | 0.26 | 0.44 |
| ToeSegmentation1 | 0.8917 | 0.7035 | 0.6944 | 0.5958 | 0.5632 | 0.5083 | 0.6708 |
| ToeSegmentation2 | 0.9353 | 0.7624 | 0.8811 | 0.8113 | 0.4348 | 0.5485 | 0.6943 |
| Trace | 1 | 1 | 1 | 1 | 1 | 0.9211 | 0.4211 |
| TwoLeadECG | 0.8551 | 0.5865 | 0.6307 | 0.5512 | 0.6262 | 0.5429 | 0.5184 |
| TwoPatterns | 0.9981 | 0.5877 | 0.8861 | 0.6994 | 0.8026 | 0.5271 | 0.7253 |
| UWaveGestureLibraryAll | 0.9905 | 0.6449 | 0.9858 | 0.9894 | 0.8058 | 0.9218 | 0.7575 |
| UWaveGestureLibraryX | 0.7011 | 0.6078 | 0.6428 | 0.6708 | 0.5054 | 0.5572 | 0.4696 |
| UWaveGestureLibraryY | 0.9839 | 0.6152 | 0.9744 | 0.96 | 0.9275 | 0.908 | 0.6198 |
| UWaveGestureLibraryZ | 0.9944 | 0.6393 | 0.9839 | 0.984 | 0.9595 | 0.9121 | 0.8527 |
| Wafer | 0.9572 | 0.4322 | 0.8668 | 0.9415 | 0.5939 | 0.8235 | 0.531 |
| Wine | 1 | 0.9477 | 0.4963 | 0.5024 | 0.5804 | 0.6778 | 0.6296 |
| WordSynonyms | 0.9687 | 0.723 | 0.9592 | 0.9387 | 0.7936 | 0.8107 | 0.8357 |
| Worms | 0.9924 | 0.803 | 0.8889 | 0.6667 | 0.7045 | 0.3333 | 0.5795 |
| WormsTwoClass | 0.9455 | 0.8 | 0.6307 | 0.6875 | 0.7531 | 0.375 | 0.4432 |
| Yoga | 0.7161 | 0.5726 | 0.6431 | 0.5919 | 0.5621 | 0.5359 | 0.5679 |

## D.2 ANOMALY DETECTION PERFORMANCE FOR UNIVARIATE TIME SERIES (CONTAMINATED TRAINING DATASET)

Table A4: AUC scores of TimeAutoML when univariate time series training datasets are contaminated with 5% and 10% anomaly samples.

| Ratio | ECG200 | ECGFiveDays | GunPoint | ItalyPowerDemand | MedicalImages | MoteStrain |
|---|---|---|---|---|---|---|
| 0% | 0.9349 | 0.9719 | 0.9093 | 0.8811 | 0.7693 | 0.9186 |
| 5% | 0.9305 | 0.9697 | 0.8994 | 0.8624 | 0.7598 | 0.9104 |
| 10% | 0.9271 | 0.9624 | 0.8902 | 0.8466 | 0.7455 | 0.9043 |
| **Decline** | **0.78%** | **0.95%** | **1.91%** | **3.45%** | **2.38%** | **1.43%** |

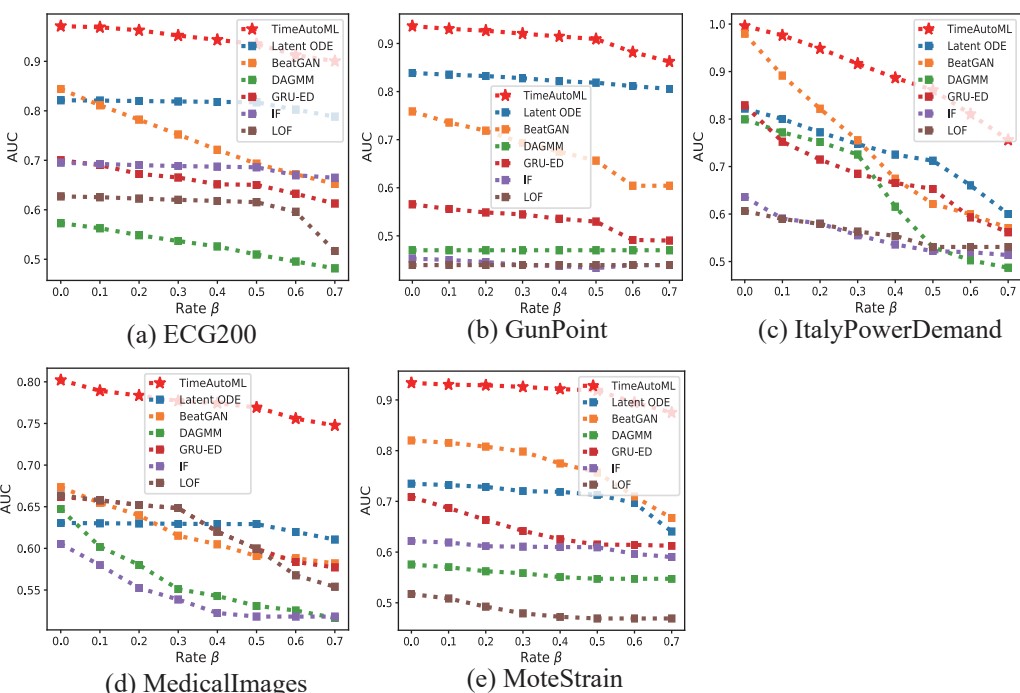

Figure A1: AUC scores of TimeAutoML and state-of-the-art anomaly detection methods on univariate datasets when irregular sampling rate $\beta$ varies from 0 to 0.7.

## D.3 ANOMALY DETECTION PERFORMANCE FOR MULTIVARIATE TIME SERIES (CONTAMINATED TRAINING DATASET)

Table A5: AUC scores of TimeAutoML when univariate time series training datasets are contaminated with 5% and 10% anomaly samples.

| Ratio | FingerMovements | LSST | RacketSports | PhonemeSpectra | Heartbeat | EthanolConcentration |
|-------|-----------------|--------|--------------|----------------|-----------|----------------------|
| 0% | 0.9643 | 0.7827 | 0.9826 | 0.8685 | 0.7703 | 0.8561 |
| 5% | 0.9554 | 0.7643 | 0.9796 | 0.8623 | 0.7604 | 0.8425 |
| 10% | 0.9388 | 0.7559 | 0.9724 | 0.8601 | 0.7527 | 0.8379 |
| **Decline** | **2.55%** | **2.68%** | **1.02%** | **0.84%** | **1.76%** | **1.82%** |

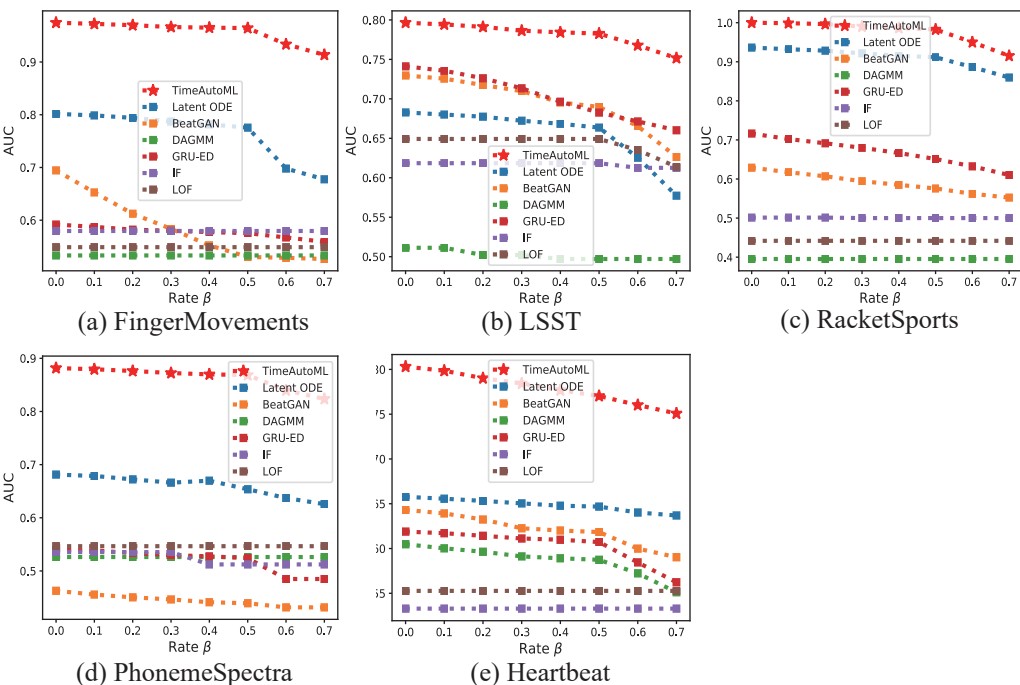

(a) FingerMovements     (b) LSST     (c) RacketSports

(d) PhonemeSpectra     (e) Heartbeat

Figure A2: AUC scores of TimeAutoML and state-of-the-art anomaly detection methods on multivariate datasets when irregular sampling rate $\beta$ varies from 0 to 0.7.

# E    APPENDIX E: ILLUSTRATION OF IRREGULAR SAMPLING

We provides an illustrative example to demonstrate how TimeAutoML remain robust against the irregularities in the sampling rates. Both regularly and irregularly sampled time series ($\beta = 0.5$) are presented in Figure A3. And for the purpose of illustration we assume the normal time series is a Sine curve. The anomalous time series is obtained via adding noise to the normal time series over a short time interval. The irregularly sampling rate is set as 0.5 in this example.

As evident from the figure, the unusual pattern of the anomalous time series preserves after the irregular sampling. And such unusual pattern appears to be different from the distortion caused by irregular sampling. Due to the special characteristics of TimeAutoML, e.g., embedded LSTM encoder and attention mechanism, it is capable of learning both the short term and long term correlations among the training time series and therefore can detect such unusual pattern even in the presence of irregularity in the sampling.

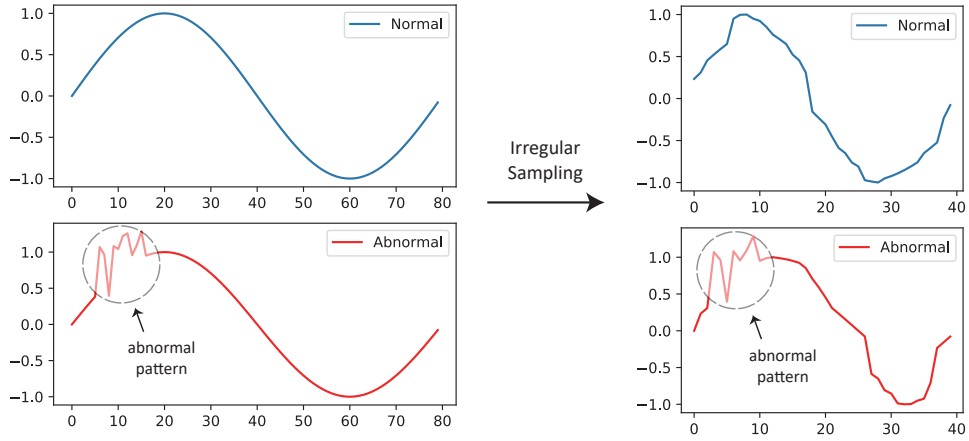

Figure A3: Irregular sampling on Sine curves.

## F   APPENDIX F: RESULTS OF TIME SERIES CLASSIFICATION

The representation generated by TimeAutoML can be used to carry out time series classification. We combine the representation generated by TimeAutoML with Gradient Boost Decision Tree (GBDT) to carry out time series classification. Followed Franceschi et al. (2019), we compared our method with four best supervised state-of-the-art model, HIVE-COTE (Lines et al., 2018), ST (Bostrom & Bagnall, 2017), EE (Lines & Bagnall, 2015), BOSS (Schäfer, 2015), which are shown in Table A6. It is worthy mentioning that both of TimeAutoML and Franceschi et al. (2019) are unsupervised time series representation learning models. Accuracy score is used as the metric to evaluate the classification performance. Besides, the results of DTW (Lei et al., 2017), which is an unsupervised model, and the results of GBDT on original input have also been posted in Table A6.

For easier comparison, we compute the average ranks of TimeAutoML and the benchmarks, which are shown in Figure A4. It is seen that our method is globally the second-best one (with average rank 2.57), only beaten by HIVE-COTE (2.42), which is a powerful ensemble method using many classifiers in a hierarchical voting structure. And we can observe that, GBDT (6.63) on original input has a poor performance, while combined with representation generated by TimeAutoML can improve the classification performance significantly, which indicating the important role of TimeAutoML.

Table A6: Accuracy scores of TimeAutoML and the state-of-the-art classification methods on UCR time series datasets when time series are regularly sampled ($\beta = 0$). Bold scores represent the best performing method.

| Dataset | TimeAutoML | Franceschi et al. (2019) | DTW | ST | BOSS | HIVE-COTE | EE | GBDT |
|---|---|---|---|---|---|---|---|---|
| ArrowHead | 0.846 | 0.829 | 0.703 | 0.737 | 0.834 | **0.863** | 0.811 | 0.629 |
| Beef | 0.8 | 0.7 | 0.633 | 0.9 | 0.8 | **0.933** | 0.633 | 0.633 |
| BeetleFly | **1** | 0.9 | 0.7 | 0.9 | 0.9 | 0.95 | 0.75 | 0.75 |
| BirdChicken | **1** | 0.8 | 0.75 | 0.8 | 0.95 | 0.8 | 0.8 | 0.9 |
| Car | 0.733 | 0.817 | 0.733 | **0.917** | 0.833 | 0.867 | 0.833 | 0.567 |
| CBF | 0.986 | 0.994 | 0.997 | 0.974 | 0.998 | **0.999** | 0.998 | 0.817 |
| Coffee | **1** | **1** | **1** | 0.964 | **1** | **1** | **1** | 0.964 |
| Computers | **0.78** | 0.628 | 0.7 | 0.736 | 0.756 | 0.76 | 0.708 | 0.58 |
| DiatomSizeReduction | **0.993** | 0.993 | 0.967 | 0.925 | 0.931 | 0.941 | 0.944 | 0.941 |
| DistalPhalanxOutlineCorrect | **0.829** | 0.768 | 0.717 | 0.775 | 0.728 | 0.772 | 0.728 | 0.728 |
| DistalPhalanxOutlineAgeGroup | **0.813** | 0.734 | 0.77 | 0.77 | 0.748 | 0.763 | 0.691 | 0.772 |
| DistalPhalanxTW | **0.719** | 0.676 | 0.59 | 0.662 | 0.676 | 0.683 | 0.647 | 0.662 |
| Earthquakes | **0.827** | 0.748 | 0.719 | 0.741 | 0.748 | 0.748 | 0.741 | 0.748 |
| ECG200 | **0.93** | 0.9 | 0.77 | 0.83 | 0.87 | 0.85 | 0.88 | 0.84 |
| ECGFiveDays | 0.979 | **1** | 0.768 | 0.984 | **1** | **1** | 0.82 | 0.787 |
| FaceFour | 0.852 | 0.875 | 0.83 | 0.852 | **1** | **0.955** | 0.909 | 0.682 |
| GunPoint | 0.993 | 0.993 | 0.907 | **1** | **1** | **1** | 0.993 | 0.84 |
| Ham | **0.771** | 0.695 | 0.467 | 0.686 | 0.667 | 0.667 | 0.571 | 0.676 |
| Haptics | 0.455 | 0.455 | 0.377 | **0.523** | 0.461 | 0.519 | 0.393 | 0.461 |
| Herring | **0.766** | 0.578 | 0.531 | 0.672 | 0.547 | 0.688 | 0.578 | 0.672 |
| ItalyPowerDemand | **0.975** | 0.925 | 0.95 | 0.948 | 0.909 | 0.963 | 0.962 | 0.966 |
| LargeKitchenAppliances | 0.859 | 0.848 | 0.795 | 0.859 | 0.765 | **0.864** | 0.811 | 0.579 |
| Lightning2 | 0.869 | **0.918** | 0.869 | 0.738 | 0.836 | 0.82 | 0.885 | 0.705 |
| Lightning7 | 0.767 | **0.795** | 0.726 | 0.726 | 0.685 | 0.74 | 0.767 | 0.616 |
| Meat | **1** | 0.95 | 0.933 | 0.85 | 0.9 | 0.933 | 0.933 | 0.9 |
| MiddlePhalanxOutlineCorrect | **0.856** | 0.814 | 0.698 | 0.794 | 0.78 | 0.832 | 0.784 | 0.801 |
| MiddlePhalanxOutlineAgeGroup | **0.656** | **0.656** | 0.5 | 0.643 | 0.545 | 0.597 | 0.558 | 0.532 |
| MiddlePhalanxTW | **0.623** | 0.61 | 0.506 | 0.519 | 0.545 | 0.571 | 0.513 | 0.552 |
| MoteStrain | 0.897 | 0.871 | 0.835 | 0.897 | 0.879 | **0.933** | 0.883 | 0.796 |
| OliveOil | **0.9** | **0.9** | 0.833 | **0.9** | 0.867 | **0.9** | 0.867 | 0.833 |
| Plane | **1** | 0.9 | **1** | **1** | **1** | **1** | **1** | 0.838 |
| ProximalPhalanxOutlineAgeGroup | **0.888** | 0.854 | 0.805 | 0.844 | 0.834 | 0.859 | 0.805 | 0.854 |
| RefrigerationDevices | 0.579 | 0.517 | 0.464 | **0.581** | 0.499 | 0.557 | 0.437 | 0.499 |
| ShapeletSim | **1** | 0.817 | 0.65 | 0.956 | **1** | **1** | 0.817 | 0.489 |
| SonyAIBORobotSurface1 | **0.965** | 0.897 | 0.725 | 0.844 | 0.632 | 0.765 | 0.704 | 0.805 |
| SonyAIBORobotSurface2 | **0.963** | 0.934 | 0.831 | 0.934 | 0.859 | 0.928 | 0.878 | 0.752 |
| Symbols | 0.849 | 0.965 | 0.95 | 0.882 | 0.967 | **0.974** | 0.96 | 0.714 |
| SyntheticControl | 0.983 | 0.983 | 0.993 | 0.983 | 0.967 | **0.997** | 0.99 | 0.883 |
| ToeSegmentation1 | 0.86 | 0.952 | 0.772 | 0.965 | 0.939 | **0.982** | 0.829 | 0.57 |
| ToeSegmentation2 | 0.877 | 0.885 | 0.838 | 0.908 | **0.962** | 0.954 | 0.892 | 0.577 |
| Trace | **1** | **1** | **1** | **1** | **1** | **1** | 0.99 | 0.8 |
| TwoLeadECG | 0.949 | **0.997** | 0.905 | **0.997** | 0.981 | 0.996 | 0.971 | 0.726 |
| Wine | **0.944** | 0.87 | 0.574 | 0.796 | 0.741 | 0.778 | 0.574 | 0.537 |
| Worms | **0.792** | 0.714 | 0.584 | 0.74 | 0.558 | 0.558 | 0.662 | 0.468 |
| WormsTwoClass | **0.844** | 0.818 | 0.623 | 0.831 | 0.831 | 0.779 | 0.688 | 0.545 |

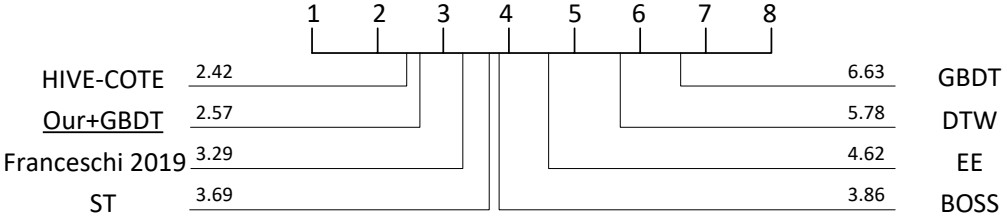

Figure A4: The average rank of our method and state-of-the-art models. Our method is an unsupervised representation learning method, combined with other classification algorithm can reach an excellent classification performance. We combine GBDT with our model in the experiment. A lower value of average rank represents a better classification performance.

## G APPENDIX G: PARAMETER SENSITIVITY

In our experiment, $\lambda_1$ and $\lambda_2$ are two weighting factors governing the trade-off among three parts in our objective in (8). We analyzed the sensitivity of $\lambda_1$ and $\lambda_2$ in ECG200 dataset (average over 10 trials) on anomaly detection task, which is shown in Figure A5. It is seen that, our approach is not very sensitive to the choice of $\lambda_1$ and $\lambda_2$. Nevertheless, a too large $\lambda_1$ or a too small $\lambda_2$ may not be friendly to the optimization of our model. Based on this observation, we set the $\lambda_1 = 0.0001$ and $\lambda_2 = 1$ in our experiment.

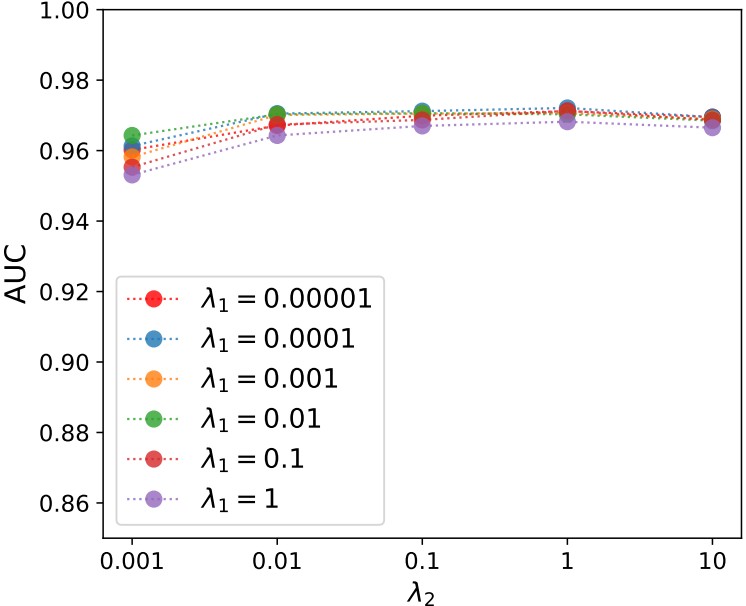

Figure A5: The sensitivity analysis about weighting factors $\lambda_1$ and $\lambda_2$, for each combination of $\lambda_1 \in \{0.00001, 0.0001, 0.001, 0.01, 0.1, 1\}$ and $\lambda_2 \in \{0.001, 0.01, 0.1, 1, 10\}$.

We also analyzed the sensitivity of Beta distribution priors $\underline{\alpha}_0$ and $\underline{\beta}_0$ in our AutoML framework, which is tested in ECG200 dataset (average over 10 trials) on anomaly detection task and the results are shown in Figure A6. It is seen that, the choice of Beta distribution priors $\underline{\alpha}_0$ and $\underline{\beta}_0$ will not

have a significant impact on performance, indicating that the our model is not very sensitive to the choice of Beta distribution priors.

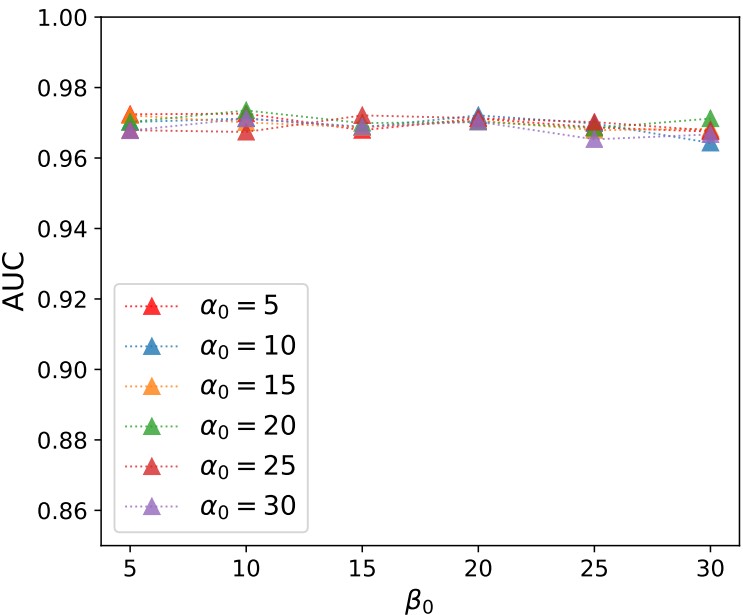

Figure A6: The sensitivity analysis about Beta distribution priors $\underline{\alpha}_0$ and $\underline{\beta}_0$, for each combination of $\underline{\alpha}_0 \in \{5, 10, 15, 20, 25, 30\}$ and $\underline{\beta}_0 \in \{5, 10, 15, 20, 25, 30\}$.

# H   APPENDIX H: MOTIVATION OF INVOLVING GAUSSIAN MIXTURE MODEL

The idea of TimeAutoML is to map the original time series into the latent space representation and the generated representation can be used to carry out many downstream machine learning tasks, such as anomaly detection, clustering, classification and so on.

We employ the GMM model to characterize the latent space representation distribution of the input data for two reasons. First, GMM is a flexible and powerful model that has been proved to be capable of approximating any continuous distribution arbitrarily well under mild assumptions. Second, once we obtain a GMM model using the training data, we can calculate the distances between an input time series and the centroids of GMM in the latent space, which are proportional to the sample energy function. Therefore, the GMM model, in combination with the sample energy function, help to characterize the level of abnormality of an input time series. The time series that is far away from the centroids of the GMM in the latent space will be deemed as an anomaly.

