# OpenReview forum: "TimeAutoML: Autonomous Representation Learning for Multivariate Irregularly  Sampled Time Series"
_ICLR.cc/2021/Conference — Reject_

### Official Review · AnonReviewer2 · 2020-10-23
**Interesting results but lack of novelty and motivations**

**Rating:** 4
**Confidence:** 4

**Review:**

I carefully read the paper and it was interesting.
The results seem promising; however, the novelty and motivations are not that satisfied.
Detailed comments are as follows.

1. Challenge 1 (Trial / Error)
- I think the first challenge (trial/error) is usual and many hyper-parameter tuning algorithms and tools are available. For instance, https://cloud.google.com/ai-platform/training/docs/using-hyperparameter-tuning
- I am not sure how the proposed method is compared with other AutoML frameworks and hyper-parameter tuning methods?
- Also, I am not sure what is "special" for AutoML for time-series technically.
- The methodology is the same with previous Thomson sampling and Bayesian optimization.

2. Challenge 2: Irregularly sampled time-series
- What do you think if we combine time-series imputation and previous time-series representation learnings?
- In that case, most time-series models can be directly applicable after imputation.
- Note that there are various imputation and interpolation methods for time-series data.
- I think this can be an important baseline to be compared with.
- Also, what is the proposed component for dealing with irregularly sampled data?

3. Challenge 3: Contrastive learning
- I am not sure why contrastive learning is good for time-series.
- And why does it guarantee suboptimal performance?
- If you have some motivations, please provide it in the revised manuscript.

4. Figure 1
- It is not easy to understand the key ideas in figure 1.
- There are too many equations but only one sentence in the caption.
- So, in the revised manuscript, it would be great to improve the readability of this figure with enough caption for the readers.

5. Motivations
- Equation (3): Can you explain why you select y_i as the representation? It is not well motivated.
- EM algorithm: Can you explain why EM algorithm is necessary for this?
- RNNs: Why the RNNs can be used for the encoder and decoders?
- Many selections in this paper have no underlying motivations even though there are many "blocks" in the proposed method.

6. Novelty
- I am not sure what is the novelty of this paper.
- As can be seen in equation (8), it seems like the proposed model are some combinations of previous methods.
- AutoML is definitely not the novel part because the authors utilize Thomson sampling and Bayesian optimization.
- It would be great if the authors can clarify the novelty of this paper.

7. Lambda 1 and Lambda 2
- How to optimize those two critical hyper-parameters?
- It would be good to add more ablation studies to check how much gain does each component in Equation (8) provide. Currently, I can only see the results without L_self.

8. Datasets
- In the title, abstract, and introduction, the authors highlight the "multi-variate" time-series.
- However, in the experiments, the authors provide the results on mostly univariate time-series dataset.
- If the proposed model is scalable, it would be great to show the results on various multi-variate and highly irregular time-series datasets.
- Here, it would be better if you use the dataset which already has missing components.

9. Fair comparison
- It is unclear how the authors optimize the hyper-parameters of other methods.
- If the proposed method is well optimized (among various hyper-parameter sets) but the baselines do not, it is not a fair comparison.

10. Computational load
- AutoML takes much more computational load.
- It would be good to quantitatively analyze the computational complexity of the proposed method compared with other methods.

11. Benchmarks
- I think the benchmarks are too limited.
- The authors should provide various "time-series" clustering and anomaly detection algorithms as the benchmarks.
- For instance, various algorithms in https://blog.statsbot.co/time-series-anomaly-detection-algorithms-1cef5519aef2.
- Note that K-mean, GMM type of things are not designed for time-series.

----------------------After reading the rebuttals-------------------------------

Thanks authors for the detailed response to my review.
Unfortunately, the responses are not satisfactory for me to increase the score.
Therefore, I stand on my initial score (4) as my final score.
The below are the reasons for this.

1. The authors failed to provide the comparison with other AutoML papers on time-series.
2. Imputation/Interpolation is a basic data preprocessing step. Also, the additional complexity for the imputation is marginal (especially compared with AutoML). Therefore, excluding the comparisons with imputation is hard to be accepted.
3. There are a bunch of self-supervised learning methods. It is unclear what is the motivation that the authors utilize "contrastive learning" as the self-supervised learning methods.
4. Also, motivations of many decisions in this paper are still missing. I think I am not the only person that raised this problem.
5. I asked for "quantitative" analyses on computational complexity. However, I cannot find the "quantitative" analyses in the revised manuscript and rebuttals for the computational complexity.

---

> ### Author Response · Authors · 2020-11-22
> **Response to Reviewer 2**
>
> Thank you very much for your constructive suggestions. We give point-to-point response as follows:
>
> (1)	The proposed work differs significantly from the hyperparameters tuning algorithms you mentioned. First, we alternatively optimize the AutoML pipeline (unsupervised representation learning pipeline) and hyperparameters, rather than just optimize the hyperparameters. There are few AutoML works addressing machine learning tasks for time series, the proposed work is the first that constructs an AutoML framework for time series unsupervised representation learning. Please notice that our emphasis is not regard designing a more efficient Thomson sampling or Bayesian optimization algorithm. Instead, our focus is on utilizing the TS + BO for alternatively constructing the AutoML pipeline and optimizing the hyperparameters.
>
> (2)	We can certainly consider imputation in the time series analysis. However, this will incur additional computational complexity. In contrast, the proposed work does not require any imputation and can achieve excellent performance in the presence of missing data.  For fair comparison, we did not consider the imputation in the experiment. Actually, in most experiments, it is seen that the performance of TimeAutoML model degrades only slightly in the presence of irregular sampling.
>
> (3)	Thank you for this comment. We did not mention that “contrastive learning guarantees suboptimal performance”. As far as we know, there are few works utilizing the contrastive methods for time series problems. The motivation is that we try to introduce a self-supervised contrastive loss to enhance unsupervised representation learning.
>
> (4)	We have updated the caption in Figure 1, which gives more details about the framework as shown in the revised version.
>
> (5)	The proposed model encodes the original time series into the latent space, and we combine the encoder hidden states h_i and reconstruction error z_i as the representation y_i. EM algorithm is then invoked to train of GMM. The reason about why we choose RNN as the encoder and decoder is that RNN are commonly used and very effective in capturing the temporal dynamics in time series data.
>
> (6)	To the best of our knowledge, there are few AutoML works addressing time series analysis, no prior works have considered AutoML framework for time series unsupervised representation learning. Moreover, a new self-supervised contrastive loss is proposed to enhance representation learning ability.
>
> (7)	We have analyzed the sensitivity about lambda 1 and lambda 2, which is detailed in Appendix G.
>
> (8)	A substantial amount of experiments have been conducted based on different multivariate time series datasets, e.g., FingerMovements, LSST, RacketSports, PhonemeSpectra, Heartbeat, and so on. More details can be found in the experiments.
>
> (9)	We have optimized the hyperparameters of the state-of-the-art methods by grid-search. The proposed TimeAutoML combines an automatic pipeline construction and hyperparameter optimization, thus achieves excellent performance.
>
> (10)	Please note that the machine learning model obtained by TimeAutoML is not necessarily of higher computational complexity than those obtained by the traditional methods. We agree that the model searching process of TimeAutoML may incur additional computational complexity than the traditional methods. In the experiments, it turns out the searching process does not take much time because the searching algorithm is quite efficient. In addition, methods such as early stopping can help speed up the training process and obtain a model more efficiently in practice.
>
> (11)	We have compared our model with state-of-the-art methods for time series analysis. For anomaly detection, we utilized BeatGAN (2019 IJCAI) and Latent ODE (2019 NIPS) as the benchmarks. For the clustering task, we utilized SPIRAL (2019 IJCAI) and DTCR (2019 NIPS) as the benchmarks.

---

### Official Review · AnonReviewer1 · 2020-10-28
**this AutoML framework integrates many existing things which makes their contributions unclear.**

**Rating:** 3
**Confidence:** 5

**Review:**

This work proposes an AutoML framework for multivariate irregularly sampled time series. To achieve this, the proposed framework integrates different modules: data-augmentation self-supervised loss (Equation 7), an anomaly detection loss (Equation 5), and a reconstruction loss. Besides, hyperparameters and model’s configuration is optimized by using AutoML (including Bayesian optimization). The model is evaluated on the well-known time-series datasets UCR and UAE.

Comments:

***motivation***  this works combines different techniques including self-supervised learning and hyperparameter optimization. But I cannot clearly find what’s their main contributions in this work since all of these techniques used in this work seems not to be new. I encourage authors to clarify their contributions formally.

***irregularly sampled ts*** since this work is for irregularly sampled time series (see title), this work should consider how to model the irregularly sampled time series. But I cannot find any related mechanism to model or deal with irregularly sampled time series in the model section. Besides, as I know, both of the datasets UCR and UAE are not standard datasets for irregularly sampled time series. And in your experiments, you construct that time series by using an irregularly sampling rate \beta. This may be problematic since it should be missing data, but not true irregularly sampled time series. There may be two different topics in the time series community.

***representation learning*** since representation learning aims to learn some good representative features that can be easily transferred to other downstream tasks. But in this work, it seems that a single feature is learned from a segment of time series. I wonder whether the representation learned from this framework can be used for other more popular downstream tasks such as classification or prediction.

Besides, since in the objective function, there are clustering loss and anomaly detection loss, I think we cannot say it is an unsupervised representation learning approach. It is more like a clustering model or an anomaly detection model. To demonstrate that the proposed framework can produce a good time-series representation, other downstream tasks such as classification or prediction need to be considered.

---

> ### Author Response · Authors · 2020-11-22
> **Response to Reviewer 1**
>
> Thank you for your comments and suggestions.  We give a point-to-point response as follows.
>
> (1)	Motivation. AutoML for time series analysis is a less-explored topic. As far as we are aware of, no prior works have considered AutoML for time series representation learning. Moreover, we propose a new self-supervised contrastive loss to enhance the model representation learning ability. These constitute two major contributions of our work.
>
> (2)	Irregularly sampled time series. What we emphasize on in this paper is a special type of irregularly sampled time series. We have made the classification in the revised manuscript. The experiments are conducted on UCR/UEA datasets with missing timestamps. More details have been given in Appendix E.
>
> (3)	Representation learning. We really appreciate your comment. The representation learned from this framework can be used for other downstream tasks. We show the classification results of TimeAutoML in Appendix F.

---

### Official Review · AnonReviewer4 · 2020-10-30
**the methodology is not well motivated.**

**Rating:** 4
**Confidence:** 4

**Review:**

This paper proposes an autonomous representation learning framework for multivariate time series with irregular sampling rates. Specifically, there are three major components proposed in the framework. 1) An AutoML solution for hyperparameters optimization under Bayesian framework is proposed to automatically seek optimal network structures and parameters. 2) Variational autoencoders based on generative approach and attention mechanism is employed to learn the semantic representation of time series with limitation of irregular sampling. 3) A sample energy function derived from Gaussian mixture model attempts to depict the level of anomaly of sliced time series.

The proposed unsupervised time series representation learning is novel in terms of AutoML methodology and contrastive learning approach. However, there are some concerns as follows:

(1)	The motivation for involving GMM model is not clearly discussed. The proposed sample energy function to denote the level of anomaly is not properly motivated as well.

(2)	There is no discussion about batching and computational complexity. Because of irregular sampling, one computational difficulty comes from the observation that times can be different for each time series in a minibatch. Furthermore, the overall AutoML pipeline without human experience suffers high computational complexity and is usually time-consuming in practice.

(3)	The experiments are not convincing. First, there is no sensitivity analysis of the hyperparameters employed in the proposed framework. Then, the datasets are not all carefully selected. In this paper, datasets are generated from normal time series datasets with a specially designed irregular sampling policy. This paper lacks experiments on irregularly sampled time series data from real scenarios. For example, PhysioNet [1] (Physionet Challenge 2012 dataset) dataset is a widely used benchmark to handle irregularly sampled time series data in many previous papers. Consequently, the performance evaluation on PhysioNet or similar datasets should be included in this paper.

[1] Silva, Ikaro et al. “Predicting In-Hospital Mortality of ICU Patients: The PhysioNet/Computing in Cardiology Challenge 2012.” Computing in cardiology vol. 39 (2012): 245-248.

---

> ### Author Response · Authors · 2020-11-22
> **Response to Reviewer 4**
>
> Thank you for your constructive comments. Per your suggestions, we have made the following modifications:
>
> (1)	We employ the GMM model to characterize the latent space representation distribution of the input data for two reasons. First, GMM is a flexible and powerful model that has been proved to be capable of approximating any continuous distribution arbitrarily well under mild assumptions. Second, once we obtain a GMM model using the training data, we can calculate the distances between an input time series and the centroids of GMM in the latent space, which are proportional to the sample energy function. Therefore, the GMM model, in combination with the sample energy function, help to characterize the level of abnormality of an input time series. The time series that is far away from the centroids of the GMM in the latent space will be deemed as an anomaly. For clarity, we have added the motivation of using GMM and the sample energy function into Appendix H.
>
> (2)	Notice that the time series dataset under investigation in this work is much smaller than the image dataset used for the experiment in the literature. We therefore use all samples in the training dataset to train our model. Please note that the machine learning model obtained by TimeAutoML is not necessarily of higher computational complexity than those obtained by the traditional methods. We agree that the model searching process of TimeAutoML may incur additional computational complexity than the traditional methods. However, in practice, methods such as early stopping can help speed up the training process and obtain a model more efficiently. In addition, TimeAutoML is a flexible framework in which we can adjust the size of the searching space according to practical needs so that tradeoffs between the complexity and the model detection performance can be achieved.
>
> (3)	The hyperparameters in our model are automatically optimized. And we analyze the sensitivity of the weighting factors (Lambda 1 and Lambda 2 in Eq (8)) and Beta distribution priors (alpha_0 and beta_0) in Appendix G. Moreover, what we emphasize on is a special type of irregular sampling pattern of time series data, we have clarified this point in the revised manuscript.

---

### Author Response · Authors · 2020-11-22
**Response to all reviewers**

  We appreciate your constructive comments and suggestions. We have revised the manuscript according to your comments, as given below:

(1)	We have revised the caption of Figure 1 to improve its readability.

(2)	We have provided the experimental results for time series classification in Appendix F.

(3)	Sensitivity analysis has been added into Appendix G.

(4)	Details about the motivation of involving GMM in our model are given in Appendix H.

  To facilitate reading, we have highlighted the revised part in blue.

---

### Decision · Program_Chairs · 2021-01-07
**Final Decision**

**Decision:**

Reject

**Comment:**

The paper introduces an AutoML method for irregular multivariate time series.   The method automates the selection of the configuration as well as the hyperparameter optimization depending on the task. A Bayesian approach handles the network structure search while VAEs + attention is used to learn  representations from irregularly sampled data. There is an additional contribution: anomaly detection via a sample energy function from a GMM on time windows.

While there is some novelty in the proposed approach, mostly in the way in which existing techniques are combined, the paper also has some limitations:
- running the framework over the set of possible models is computationally intensive; in their response, the authors indicate the search space can be constrained, however, doing so would also decrease the performance; in AutoML, added complexity cannot be avoided, but there is no notion of how much longer it takes to find suitable models compared to taking off-the-shelf methods.
- although the paper is geared towards irregularly sampled time series, there are no experiments where the data is naturally irregularly samples; artificially introduced patterns are no substitute for this; (PhysioNet, as suggested by one of the reviewers or MIMIC III both have this type of data and are frequently used in benchmarks)
- AutoML is presented as a general framework, but mostly handles clustering and anomaly detection;  unclear of how useful it would be for forecasting or regression; classification realists are shown in Appendix F against simple baselines (GRU-D is not considered, for instance) and even so AutoML does not achieve state of the art results in half of the cases